# Bispyrrolidinoindoline Epi(poly)thiodioxopiperazines (BPI-ETPs) and Simplified Mimetics: Structural Characterization, Bioactivities, and Total Synthesis [note 1]

**DOI:** 10.3390/molecules27217585

**Published:** 2022-11-04

**Authors:** Claudio Martínez, Patricia García-Domínguez, Rosana Álvarez, Angel R. de Lera

**Affiliations:** CINBIO, ORCHID Group, Departmento de Química Orgánica, Universidade de Vigo, 36310 Vigo, Spain

**Keywords:** bispyrrolidinoindoline epi(poly)thiodioxopiperazine alkaloids, isolation, structural elucidation, total synthesis, bioactivities, synthetic analogs

## Abstract

Within the 2,5-dioxopiperazine-containing natural products generated by “head-to-tail” cyclization of peptides, those derived from tryptophan allow further structural diversification due to the rich chemical reactivity of the indole heterocycle, which can generate tetracyclic fragments of hexahydropyrrolo[2,3-*b*]indole or pyrrolidinoindoline skeleton fused to the 2,5-dioxopiperazine. Even more complex are the dimeric bispyrrolidinoindoline epi(poly)thiodioxopiperazines (BPI-ETPs), since they feature transannular (poly)sulfide bridges connecting C3 and C6 of their 2,5-dioxopiperazine rings. Homo- and heterodimers composed of diastereomeric epi(poly)thiodioxopiperazines increase the complexity of the family. Furthermore, putative biogenetically generated downstream metabolites with C11 and C11’-hydroxylated cores, as well as deoxygenated and/or oxidized side chain counterparts, have also been described. The isolation of these complex polycyclic tryptophan-derived alkaloids from the classical sources, their structural characterization, the description of the relevant biological activities and putative biogenetic routes, and the synthetic efforts to generate and confirm their structures and also to prepare and further evaluate structurally simple analogs will be reported.

## 1. Introduction 

Dimeric bispyrrolidinoindoline epi(poly)thiodioxopiperazines (BPI-ETPs) are a family of highly complex natural products that biogenetically derive from dioxopiperazines formed by the double intramolecular condensation of dipeptides containing tryptophan and an additional amino acid, followed by a variety of structural modifications [1,2,3]. The monomeric units contain cyclic dipeptide (CDP) substructures fused to pyrrolidinoindoline cores and feature transannular (poly)sulfide connections between C3 and C6 (for the numbering indicated in Figure 1, see [4]) of their 2,5-dioxopiperazine rings [1]. Relevant features of these privileged structures include the conformational constraint, which has been linked to their greater stability and conformational rigidity and, therefore, higher resistance to protease degradation than acyclic counterparts, as well as their ability to cross the intestinal barrier and the blood–brain barrier [5]. The ability to mimic preferential peptide conformations, with two hydrogen bond donor and acceptor sites, favors interactions with putative biological targets. Thus, their pharmacological potency is boosted when compared with the monomeric unit through multipoint interactions on chemical space with their biological targets [6,7]. 

More than 100 alkaloids containing at least a unit of indole-derived [8] pyrrolidinoindoline epi(poly)thiodioxopiperazine (PI-ETP) [9] have already been isolated from terrestrial and marine fungi (about a third, from marine sources) [10,11,12,13], including species of the *Leptosphaeria*, *Chaetomium*, [14] *Tilachlidium*, *Verticillium*, *Gliocladium*, *Aspergillus* sp. and *Penicillium* genera [1,15,16], but they have also been obtained from bacteria (*Streptomyces* species) [17], plants, and even animals [5,13,18]. Homo- and heterodimeric BPI ETPs compose about half of the family. 

(+)-Chaetocin A (**1**, Figure 1A) [19] and (+)-verticillin A (**8**, Figure 2) [20] were first discovered in 1970, although antibacterial monomeric (−)-gliotoxin (**7**, see Figure 1B) had been isolated more than 30 years before [21,22]. Relative to the monomers, the homo- and heterodimeric BPI ETP family members feature a greater level of structural diversity and complexity since the characteristic dioxopiperazine scaffold is bridged by a variable number (1 to 4) of sulfur atoms. Nondimeric family members connected at C3 through the indole nitrogen of either simple indole units or epi(poly)thiodioxopiperazines will not be covered. 

Biogenetic generation of CDPs in nature usually involves either nonribosomal peptide synthetases (NRPSs) in fungi [23,24] or cyclodipeptide synthases (CDPSs) in bacteria [25,26,27]. Whereas fungi generate DKP monomers by the activation of the free amino acids through adenylation by the action of NRPSs, bacteria use cyclodipeptide synthetases (CDPSs) and employ aminoacyl-tRNAs (aa-TRNAs) as substrates [26,27,28,29,30]. 

The diversification of the skeleton is further achieved through the biogenetically controlled action of their tailoring enzymes, which are usually found in dedicated biosynthetic gene clusters [31]. Whereas, for the bispyrrolidinoindoline dioxopiperazine, family various types of oxidoreductases, hydrolases, transferases, and ligases have been isolated and structurally characterized [32], for the BPI-ETP family, these chemical modifications are rather limited. In addition to the role of Cyt P450 oxidation enzymes [33], as well as enzymatic dimerization [34,35], only hydroxylation/dehydroxylation processes leading ultimately to the modulation of the oxidation level of the pyrrolidinoindoline and DKP scaffolds and the amino acid side chains have been noted in some members of the family of dimeric alkaloids. In contrast, enzymes responsible for structural modifications of the DKP scaffold in simpler bispyrrolidinoindoline dioxopiperazine alkaloids have been characterized in several biogenetic gene clusters of fungal and microbial secondary metabolites [1,5,29,36,37,38,39,40,41,42].

The conformationally constrained DKP scaffold of dimeric dioxopiperazines is currently considered as privileged structures [43], given their ability to interact with several receptors, which may account for the diverse biological activities reported for this family of natural products. In addition, the reactivity of compounds with di(poli)sulfide bridges has been associated with a variety of biological activities, including protein cross-linking through the reaction of the disulfide bond with cysteine residues and the inactivation of thiol-containing proteins, generation of reactive oxygen species (ROS) via redox cycling, or ejection of zinc ions from some proteins [10,11,12,44,45,46].

Recent reviews on isolation and bioactivities of dimeric bispyrrolidinoindoline DKPs lacking (poly)sulfide bridges [1,13,32] and analogs with the same structural unit have been published [3,13,47,48,49,50], but their biogenesis and total synthesis, as well as that of simple analogs, have not been covered in depth. Nonlipopeptide fungi-derived peptide antibiotics reported since 2000 have also been reviewed [51].

The present revision collects dimeric natural products with BPI-ETP skeletons (formally with epi(poly)thiopyrazino[1’,2’:1,5]pyrrolo[2,3-*b*]indole-1,4-dione substructures) that are derived from tryptophan-containing peptides, their traditional isolation, structural elucidation and putative biogenesis, and the few approaches that achieved the challenging synthesis of these complex dimeric structures. The reported members of this family of alkaloids will be discussed, with an arbitrary distribution based on the relative configuration and identity of the second amino acid component of the DKP scaffold. In addition, structurally simple analogs displaying some key features of the natural products, and their biological activities in comparison with those of the inspiring targets, will also be described.

## 2. Isolation, Structural Elucidation, and Biological Activities of Tryptophan-Derived BPI-ETP Alkaloids 

### 2.1. Structural Elucidation

The general protocols for the structural characterization of tryptophan-derived BPI-ETP alkaloids start with the MS data obtained by high-resolution fast atom bombardment mass spectrometry (HRFABMS); the analysis of the pseudomolecular ion, which allows for suggesting the general molecular formula of C_x_H_y_N_z_O_v_S_w_; and the number of double bond equivalents (DBE) or degrees of unsaturation. Since in the MS spectra, sulfur-containing [M + 2] isotope peaks are observed by LC–MS profiling, the number of sulfur atoms in the (poly)sulfide bridge(s) can be deduced from the analysis of mass spectral data combined with chemical derivatization. In particular, disulfide reduction with NaBH_4_ and treatment with CH_3_I to provide derivatives of the natural product with methylsulfanyl groups has been thoroughly used. Disulfide cleavage and desulfuration can also be induced upon treatment with 5% KOH in dioxane-H_2_O and aluminum amalgam [20,52]. Analysis of FABMS fragments in the MS spectra of these NPs and their derivatives also provides information regarding the location of these substituents on each half of the dimeric structure. 

The presence of certain functional groups (most notably, hydroxyl, amino, and amide) and of aromatic rings can also be associated with characteristic bands on the IR spectra of these alkaloids. 

^1^H and ^13^C NMR spectroscopic analysis by DEPT and ^1^H-^1^H and ^1^H-^13^C correlation spectroscopy experiments and comparison of spectroscopic data with those of related compounds allow for identifying fragments and suggesting their connectivity. The location of the sulfur-containing functional group in the corresponding dioxopiperazine ring is usually based on HMBC correlations, aided by the comparison of the NMR signals of the synthetic methylsulfanyl derivatives with those of the natural product. Extension of the HMBC correlation sequences from the amino-acid-containing regions to neighboring atoms through the ^1^H NMR resonances could indicate the presence of structural fragments, which must be connected, taking into consideration ROESY correlations, thus suggesting the relative configurations. 

^13^C NMR spectroscopic analysis in this series also provides putative resonance signals for the amide carbonyl (δ_C=O_ around 177 ppm), and the information combined with the DBE confirms the polycyclic nature of the natural product. 

Analysis of the corresponding derivatives (acetates, methylsulfanyl, …) by DEPT and ^1^H-^1^H and ^1^H-^13^C COSY experiments allows for carrying out the comparison with other family members and identifying the functional groups present in the natural product. As indicated above, partial desulfuration upon treatment with PPh_3_ at ambient temperature generates the corresponding disulfides /sulfides (**I** and **II**, n = 1 or 2, Figure 1) [53,54,55,56], but it can be further modulated upon careful heating to 60 °C to afford the C3-connected indole derivatives (**IV** and **VI**, Figure 1). 

Optical rotation at a fixed wavelength, optical rotatory dispersion (ORD), and electronic circular dichroism (ECD) of the natural products, in particular, combined with theoretical simulation of the spectra, allow the determination of their absolute configuration. In these sulfur-containing BPI ETPs, the negative band at λ_max_ 271 nm in the CD spectrum of the natural product has been assigned to the S→CO charge transfer, which also allows for suggesting the configuration, and further corroborates it by NOE experiments [57]. In addition, the configuration at C-3/C-3’ can be proposed based on the CD Cotton effect at λ_max_ 248 nm corresponding to the π→π* transition [58]. 

Pioneering X-ray diffraction analysis on these natural products [19,59], aided by the anomalous dispersion effect of sulfur, has been used to determine the absolute configuration.

### 2.2. BPI-ETP Alkaloids 

Figure 1, Figure 2, Figure 3, Figure 4, Figure 5, Figure 6, Figure 7 and Figure 8 collect the reported BPI-ETP natural products. This collection of alkaloids illustrates the unmatched power of nature to achieve structural diversity not only connecting Trp with other amino acids (Ala, Ser, Val, and Thr), but also altering the stereogenicity of the chiral centers, the configuration of the dioxopiperazine-containing stereocenters, the nature and location of substituents at several positions, and the number of sulfur atoms on the epi(poly)thiodioxopiperazine ring.

#### 2.2.1. Chaetocin and Analogues

(+)-Chaetocin A (**1**, Figure 1) [19] and (+)-verticillin A (**8**, Figure 2) [20] were the first members of the group reported. 

(+)-Chaetocin A (**1**) was first isolated from the fermentation broth of *Chaetomium minutum* [19,59], and more recently, it was obtained from the marine-derived fungus *Nectria inventa* [60]. 

Structural elucidation of (+)-chaetocin A (**1**) started with the ^1^H and ^13^C NMR spectroscopic analysis of solutions in (CD_3_)_2_SO, which revealed the presence of the *N*-Me group (δ 3.02 ppm), a methylene (δ 2.92 and 3.64 ppm, *J* = 15 Hz), and an A_2_X system (δ 4.18 and 5.80 ppm). IR bands for hydroxymethyl substituents (ν_max_ 3550, 3430 cm^−1^) were compatible with the absence of the coupling signal for the CH_2_OH group (*J* = 5.5 Hz) observed in ^1^H NMR upon treatment with D_2_O and the formation of the corresponding acetates with acetic anhydride and pyridine [19]. Structural similarity to monomeric (−)-gliotoxin (**7**, Figure 1B), which at that time was known to be produced by species of the fungi *Aspergillus*, *Penicillium*, and *Alternaria* [61], was noted. Signals at ν_max_ 1690, 1680, and 1660 cm^−1^ in the IR spectrum and the C_30_H_28_N_6_O_6_S_4_ molecular formula were also an indication of a dithiodioxopiperazine ring on a dimeric structure. X-ray diffraction analysis [19,59] allowed for also determining the absolute configuration. The two epidithiodioxopiperazine units in (+)-chaetocin A (**1**) have almost exact C2 symmetry with torsion angles around C_3_-C_3’_ bond corresponding to a staggered conformation. The differences in the sign of the -C-S-S-C- torsion angle with respect to monomeric structures including that of (−)-gliotoxin (**7**, Figure 1B) [61] indicated the opposite configuration of the stereocenters at the epidithiodioxopiperazine fragment in (+)-chaetocin A (**1**) [59].

A variety of pharmacological activities of (+)-chaetocin A (**1**) have been reported (for a recent revision, see [13]), starting with the antibacterial and cytostatic effects described following its isolation, with MIC values of 0.01 and 0.001 µg/mL for *Staphylococcus aureus* and penicillin-resistant *S. aureus*, respectively. In addition, a 50% reduction of mouse mast cell P-815 at a concentration of 0.013 mg/L was observed. The acute toxicity was determined as DL_50_ = 1.7 mg/kg after intraperitoneal injection [19]. Additional toxicity studies using ICR male mice revealed extensive fibrinous peritoneal adhesion and focal liver cell necrosis when using high doses of (+)-chaetocin A (**1**) (5.0 mg/kg). Less severe peritoneal adhesion was also determined on mice treated with lower amounts (from 0.6 to 2.5 mg/kg) of the natural product [62].

After screening 3000 compounds for inhibitory activity towards recombinant *Drosophila melanogaster* SU(VAR)3–9 protein [63], (+)-chaetocin A (**1**) was reported as an inhibitor of some members of the SU(VAR)3–9 histone lysine methyl transferase (HKMT) family of epigenetic enzymes (vide infra) [64,65], which includes, among others, SUV39H1, G9a, DIM-5, GLP, and ESET [66,67].

(+)-Chaetocin A (**1**) was also found to bind (Kd of 16.8 µM) and inhibit Hsp90 and also to degrade client proteins (EGFR, p-EGFR, Akt, p-Akt, and cyclin D1) at a similar dose than (+)-chetracin B (**27**) or HDN-1 (Figure 4). The level of lysine methyl transferase SUV39H1 declined upon treatment with these compounds, which also suggested the activity of (+)-chaetocin A (**1**) as an inhibitor of SUV39H1 [68].

The mechanistic rationale for the activity of (+)-chaetocin A (**1**) as an inhibitor of HKMTs has more recently been clarified [56,69,70,71]. Studies on the inhibition of HKMT G9a by (+)-chaetocin A (**1**) and some functionally related analogues revealed that only the epidithiodioxopiperazine core was required for the reported inhibition of the SU(VAR)3–9 class of HKMTs [63]. Such inhibitory effects are time dependent and irreversible in the absence of DTT, which is consistent with nonspecific protein denaturation. Covalent mix disulfide adduct formation with the protein was moreover demonstrated by MS analysis. Thus, the high toxicity of the ETP class of compounds, and their extensive off-target effects on a variety of structurally, functionally, and evolutionarily unrelated enzymes was proposed to be the result of diverse changes in histone modifications along the apoptotic process [70,71].

**Figure 1 molecules-27-07585-f001:**
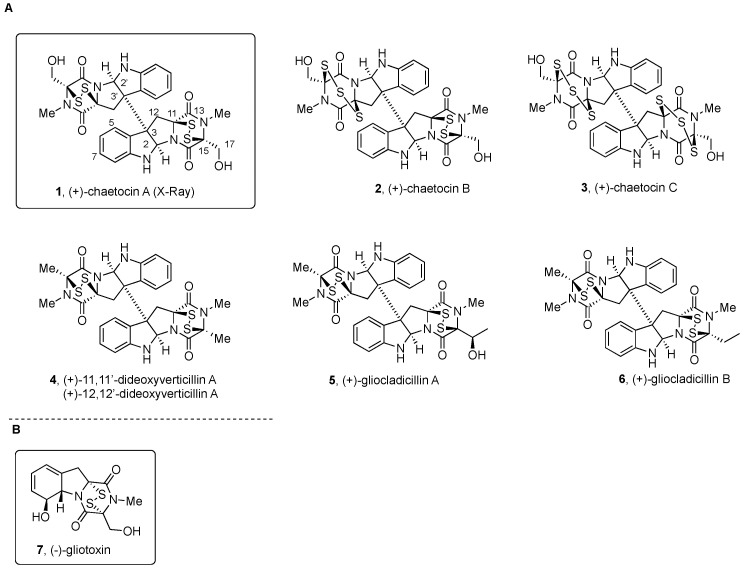
(**A**) Symmetrical and nonsymmetrical BPI ETP alkaloids derived from Trp, Ala, Ser, and Thr; (**B**) Structure of (−)-gliotoxin (**7**).

(+)-Chaetocin B and (+)-chaetocin C (**2** and **3**, respectively, Figure 1) were isolated, together with (+)-chetracin A (**27**, Figure 4) [62] and 12α,12α’-dihydroxychaetocin [19] or melinacidin IV (**13**) (first named as 11α,11α’-dihydroxychaetocin, Figure 2) [72,73], from *Chaetomium virescens* var. *thielavioideum* NHL 2827 [62]. The structures of **2** and **3** were determined to be analogous to those of (+)-chetracin A (**27**, Figure 4) previously obtained from *C. nigricolor* and *C. retardatum* [62].

Comparison of the NMR data of (+)-chaetocin B (**2**) with those of symmetrical (+)-chaetocin A (**1**) revealed the similarities on one-half of the molecule, and differences on the second half were attributed to the number of sulfur atoms in the structure. FAB MS/MS analysis on the acetate suggested (+)-chaetocin B (**2**) to be the sulfur homologue of (+)-chaetocin A (**1**) containing one trisulfide bridge. ^1^H NMR analysis of solutions of **2** in toluene-d_8_ indicated the presence of two conformers of the trisulfide fragment, with a lower percentage of the minor conformer being present at higher temperatures. The absolute configuration was proposed to be the same as that of (+)-chaetocin A (**1**) [74] by comparison of their CD spectra [62].

The same spectroscopic and chemical analysis led to suggest for the minor product (+)-chaetocin C (**3**, Figure 1) the homodimeric structure with one trisulfide bridge on each dioxopiperazine fragment [62].

Antimicrobial activities of these natural products and some derivatives with chemical modifications on the disulfide/trisulfide bridges (monosulfide, dethiomethylthio, dethioacetylthio, and dethiothiol derivatives) were then studied. Antimicrobial activities against *Staphylococcus aureus* FDA 209P, *E. coli* NIHJ, and *Saccharomyces cerevisiae* ATCC 9763 were examined using the dilution method on agar plates. None of these compounds were active against Gram-negative bacteria *E. coli* NIHJ or *Saccharomyces cerevisiae* ATCC 9763 fungi, but were found to be very active against Gram-positive bacteria *S. aureus* FDA 209P. (+)-Chaetocin B (**2**) and (+)-chaetocin C (**3**, Figure 1) with one and two trisulfide bridges, respectively, showed the highest activity (MIC of 0.05 and 0.05 µg/mL, respectively), followed by (+)-chaetocin A (**1**, 0.1 µg/mL) and (+)-chetracin A (**27**, Figure 4, 0.39 µg/mL), whereas the acyclic sulfur derivatives showed no activity [62]. In addition, strong cytotoxicity against HeLa cells (IC_50_ values from 0.02 to 0.07 µg/mL) was found for the three natural products, namely, (+)-chaetocin B and (+)-chaetocin C (**2** and **3**, respectively, Figure 1) and (+)-chetracin A (**27**, Figure 4) [62,75].

(+)-12,12’-Dideoxyverticillin A (**4**, Figure 1; first named (+)-11,11’-dideoxyverticillin) and (+)-12’-deoxyverticillin A (**5**, Figure 1; first named (+)-11’-dideoxyverticillin) were isolated, together with (+)-verticillin A (**8**, Figure 2) [20], from marine-derived fungi *Penicillium* sp. obtained from the surface of the Caribbean Chlorophyta *Avrainvillea longicaulis* collected in 1996 near Sweetings Cay, Bahamas [76]. 

The analysis of the UV, IR, EIMS data, and NMR spectra of (+)-12,12’-dideoxyverticillin A (**4**) indicated for the natural product the presence of a disulfide-bridged dioxopiperazine structure similar to that previously described for (+)-verticillin A (**8**, Figure 2) [20]. The relative configuration was proposed based on NOE experiments. The *S* absolute configuration of the stereocenters of the dioxopiperazine ring was assigned [76] after analysis of the CD data and specific rotation values and their comparison with those of (+)-chaetocin A (**1**, Figure 1) and synthetic diacetate [19,73].

(+)-12,12’-Dideoxyverticillin A (**4**), and also (+)-12’-deoxyverticillin A (**9**, Figure 2), showed potent in vitro cytotoxicity against HCT-116 human colon carcinoma (IC_50_ = 30 ng/mL) [76].

(+)-12,12’-Dideoxyverticillin (**4**, Figure 1) has more recently been isolated from the fungus *Shiraia bambusicola* that inhabits bamboo and is used in Chinese medicinal tradition to treat psoriasis [77]. It was further characterized as a growth factor receptor tyrosine kinase inhibitor with potent antitumor activity in a cell-free ELISA against EGFR, with an IC_50_ value of 0.136 nM. For VEGFR-1/Flt-1 (vascular endothelial growth factor receptor-1/fms-like tyrosine kinase-1) and HER2/ErbB-2, the inhibitory activity was weaker (IC_50_ value in the micromolar range). It was also found to inhibit the proliferation of four human breast cancer cell lines (MDA-MB-468, MCF-7, MDA-MB-435, and MDA-MB-231) with IC_50_ values from 0.13 to 0.28 µM. It also showed remarkable in vivo efficacy against mice sarcoma 180 and hepatoma 22 (with inhibition rates of 45.0 to 72.4%) after daily i.p. administration of 0.5 or 0.75 mg/kg [77]. Additional activities of (+)-12,12’-dideoxyverticillin A (**4**) have been recently reviewed [13].

(+)-Gliocladicillin A (**5**) and (+)-gliocladicillin B (**6**) (Figure 1) were isolated, together with (+)-12,12’-dideoxyverticillin (**4**) from the *Cordyceps*-colonizing fungus *Gliocladium* sp. collected at high altitude (3500 m) on the Qinghai–Tibetan plateau, where it parasitizes dead caterpillars of the moth *Hepialus* spp. [78]. The presence of a 2-hydroxyethyl unit instead of the methyl group of (+)-12,12’-dideoxyverticillin (**4**) allowed the structural assignment to (+)-gliocladicillin A (**5**). Similarly, the structure of a 17-deoxy analogue was assigned to (+)-gliocladicillin B (**6**). (+)-Gliocladicillin A (**5**) was also isolated after following a bioactivity-directed fractionation of the organic extract of two filamentous fungi of the Bionectriaceae, strains MSX 64546 and MSX 59553, from the Mycosynthetix library [79], together with (+)-Sch 52900 (**18**, Figure 3), (+)-verticillin A (**8**, Figure 2), (+)-gliocladicillin C or C77 (**16**, Figure 2), (+)-Sch 52901 (**19**, Figure 3), and (+)-12’-deoxyverticillin A (**9**, Figure 2) [79]. 

Antitumor effects at GI_50_ values ranging from 0.1 to 0.5 µg/mL in HeLa, HepG2, and MCF7 human cancer cell lines were determined using the MTT assay for (+)-gliocladicillin A (**5**) and (+)-gliocladicillin B (**6**), and tumor cell apoptosis was shown to be induced in a time- and dose-dependent manner. (+)-Gliocladicillin A (**5**), (+)-gliocladicillin B (**6**), and (+)-12,12’-dideoxyverticillin (**4**, Figure 1) caused G_2_/M cell cycle arrest and induced apoptosis in HeLa cells by activating multiple signaling pathways. They were found to activate caspase-8, and also effector caspase-3, which cleaves substrates such as PARP, leading to cell death. In addition, they upregulated the expression of p53, induced a change in the Bax/Bcl-K_L_ ratio, and activated caspase-9 through the intrinsic pathway. In vivo activities in a C57BL/6J xenograft mice model were also reported for (+)-gliocladicillin A (**5**) and (+)-gliocladicillin B (**6**) (and also (+)-12,12’-dideoxyverticillin (**4**, Figure 1)), with promising inhibitory effects for the population growth of melanoma B16 cells [78].

Symmetrical dimer (+)-verticillin A (**8**, Figure 2) was first reported as a metabolic product of the *Verticillium* sp. strain TM-759, an imperfect fungus isolated from a basidiocarp of *Coltricia cinnamomea* (*Polystictus cinnamomeus*) [20,52]. The positive Cotton effect observed at 272, 307, and 375 nm was assigned to the disulfide chromophore, and its presence was confirmed by an *m/z* ion of 64 due to the loss of S_2_, whereas the analysis of NMR data of the derivatives that resulted from the disulfide cleavage and desulfuration allowed for proposing the gross structure of (+)-verticillin A (**8**) [20,52]. (+)-Verticillin A (**8**) was further identified as a member of the BPI ETP family after comparison of the ^1^H, ^13^C, and DEPT NMR data with those of (+)-chaetocin A (**1**, Figure 1). Since the CD data were of opposite sign than those of monomeric alkaloids, such as (−)-gliotoxin (**7**, Figure 1), which have the *R* configuration on the two asymmetric centers of the dioxopiperazine ring, the configuration of the hydroxyl-containing stereogenic centers was assigned as *S*. The structural analogy to (+)-chaetocin A (**1**, Figure 1) was also based on the IR study, which showed the hydroxyl groups to be engaged in hydrogen bonding interactions with the carbonyl oxygens. The absolute configuration could also be confirmed by the analysis of the Cotton effect at 223 nm ([θ] = +50,000) for the monobenzoate derivative, which allowed for assigning the α configuration to the hydroxyl-group-containing stereocenters [20,52].

The more soluble acetate derivative of (+)-verticillin A (**8**, Figure 2) [20] showed antimicrobial activity against Gram-positive bacteria and mycobacteria, but not against Gram-negative bacteria and fungi. Cytotoxic activity against HeLa cells (ED_50_ = 0.2 µM) was also reported [20]. In vivo antitumor activity of synthetic (+)-verticillin A acetate was observed in an Ehrlich ascites carcinoma mice model [20], and a value of LD_50_ = 7.6 mg/kg for acute toxicity was determined 10 days after injection.

(+)-Verticillin A (**8**) was later isolated from wheat solid-substrate fermentation of *Gliocladium roseum* 1A, together with (+)-gliocladine A (**25**, Figure 4), (+)-gliocladine B (**26**, Figure 4), (+)-12’-deoxyverticillin A (**9**, Figure 2), (+)-Sch 52900 (**18**), and (+)-Sch 52901 (**19**, Figure 3), and found to display cytotoxic activity against a panel of human cancer cell lines, with IC_50_ values of 20 to 370 nM. In addition, they also were effective nematicidal agents towards *Caenorhabditis elegans* and *Panagrellus redivivus* (vide infra) [80].

The selective inhibition of the histone lysine methyl transferase (HMTases) SUV39H1, SUV39H2, G9a, GLP, NSD2, and MLL1, and the promotion of chromatin remodeling might explain the anticancer activity of (+)-verticillin A (**8**), which was considered a potential candidate for overcoming colon carcinoma and pancreatic ductal adenocarcinoma (PDAC) cell resistance [81,82]. 

(+)-Verticillin A (**8**, Figure 2) was more recently shown [83] to act as a sensitizer of human colon carcinoma cells to TRAIL-induced apoptosis (for a recent revision, see) [13].

(+)-12’-Deoxyverticillin A (**9**, Figure 2) and (+)-12,12’-dideoxyverticillin A (**4**, Figure 1) were isolated together with (+)-verticillin A (**8**, Figure 2) from a marine-derived fungus *Penicillium* sp., obtained from the surface of the Caribbean Chlorophyta *Avrainvillea longicaulis* [76].

The UV, IR, EIMS data, and NMR spectra (+)-12’-deoxyverticillin A (**9**) closely resembled those of (+)-verticillin A (**8**) and (+)-12,12’-dideoxyverticillin A (**4**, Figure 1), except that the ^1^H and ^13^C NMR spectra revealed the presence of slightly different dioxopiperazine subunits, one of them assigned to that present in (+)-verticillin A (**8**) and the other to the deoxy analog. The relative configuration was assigned based on comprehensive NOE spectral data. Treatment with NaBH_4_ and MeI afforded the tetrakismethylsulfanyl derivative, thus probing the presence of the epidithiodioxopiperazine functionality [76].

(+)-12’-Deoxyverticillin A (**9**) and (+)-12,12’-dideoxyverticillin A (**4**) showed potent in vitro cytotoxicity against HCT-116 human colon carcinoma (IC_50_ = 30 ng/mL) [76].

Synthetic analog G266 (**10**, Figure 2B) was tested in vitro on a panel of human breast cancer cell lines (MDA-MB-231, MDA-MB-468, MCF-7, ZR-75-30, BT474, BT549, SK-BR-3, T47D, and HBL100) [84]. G266 (**10**) suppressed the proliferation of the nine cancer cell lines with a mean IC_50_ value of 48.5 nmol/L (cf. adriamycin IC_50_ of 170.6 nmol/L), triggering autophagy and caspase-dependent apoptosis in a similar manner as (+)-12,12’-dideoxyverticillin (**4**, Figure 1) [84].

(+)-Melinacidin II, (+)-melinacidin III, and (+)-melinacidin IV (**11**, **12,** and **13**, respectively, Figure 2) were obtained from *Acrostalagmus cinnabarinus* var. *melinacidinus* [72,73]. Melinacidin IV (also known as (+)-12α,12α’-dihydroxychaetocin **13**, first named (+)-11α,11α’-dihydroxychaetocin) [19] was also isolated from *Westerdykella reniformis* sp. nov, together with (+)-chetracin B or HDN-1 (**28**, Figure 4) [85]. (+)-12α,12α’-Dihydroxychaetocin or melinacidin IV (**13**) was previously isolated from *Verticillium tenerum* [74]. Together with (+)-chetracin A (**27**, Figure 4), (+)-12α,12α’-dihydroxychaetocin or melinacidin IV (**13**, Figure 2) was isolated from *Chaetomium* sp. in the course of studies on mycotoxin production by the fungi of the species *Chaetomium abuense* Lodha and *C. retardatum* Carter and Khan [62,75]. After isolation from the first source [72], the key structural units, namely, the dioxopiperazine moiety and the disulfide connector, were determined to be present [73]. Comparison of the spectroscopic data of (+)-melinacidin IV (**13**, Figure 2) with those of (+)-12α,12α’-dihydroxychaetocin confirmed that they were the same structure.

Relative to (+)-melinacidin IV or (+)-12α,12α’-dihydroxychaetocin (**13**), (+)-melinacidin III (**12**, Figure 2) and (+)-melinacidin II (**11**, Figure 2) differ by the nature of the substituents at C17,C17’ and C12,C12’ positions, with melinacidin III (**12**) containing two hydroxymethyl (C17,C17’) and one hydroxyl (C12) group, and melinacidin II (**11**) one hydroxymethyl, a methyl (C17,C17’) and a hydroxyl substituent (C12 or C12’) at each subunit. The similarity of the CD data of melinacidins with those reported for (+)-verticillin A and (+)-verticillin B (**8** and **14**, respectively, Figure 2), and for (+)-melinacidin IV or (+)-12α,12α’-dihydroxychaetocin (**13**, Figure 2) [19,74], led to proposing the same *S* configurations for the stereocenters at the dioxopiperazine ring. Based on biogenetic considerations, the configuration of the stereocenters with hydroxyl groups at C12 of (+)-melinacidin II and (+)-melinacidin III (**11** and **12**, Figure 2) was likewise considered to be *S* [73].

(+)-Verticillin B (**14**) and (+)-verticillin C (**15**, Figure 2), also isolated from the same *Verticillium* sp. strain TM-759 source [52], were determined to share the core dimeric structure of (+)-verticillin A (**8**) but differed in the identities of the terminal amino acids of the dioxopiperazine rings [86]. (+)-Verticillin B (**14**) was later isolated from a terrestrial strain of *Gliocladium catenulatum* along with two analogues [87] and from an extract of *Nectria inventa*, collected from a sediment obtained below 600 m in Monterey Bay, CA [60].

The structures of (+)-verticillin B (**14**) and (+)-verticillin C (**15**) could be determined by comparison of their NMR data with those of (+)-verticillin A (**8**, Figure 2). For the former, the nonsymmetrical structure was assigned based on signals for a methyl group at δ 1.75 ppm and a CH_2_OH group at δ 4.52 ppm. For the latter, the presence of one additional sulfur atom suggested a trisulfide moiety in one subunit. The determination of the degradation products when compared with the units of congeners (+)-verticillin A (**8**) and (+)-verticillin B (**14**) led to proposing structure **15** for (+)-verticillin C. The absolute configuration was assigned by analogy with the other members of the group [52,57,86,88].

The trypanocidal potential of (+)-verticillin B (**14**) (IC_50_ value of 0.007 µM against *T. brucei*; IC_50_ < 0.6 µM for Jurkat cells) and (+)-chaetocin A (**1**, Figure 1) (IC_50_ of 0.002 µM against *T. brucei*; IC_50_ < 0.6 µM Jurkat cells) was reported, although with low selectivity index. The results confirmed the important role of the disulfide bond in the nanomolar activity against *T. brucei*, although a substantial toxicity to mammalian cells was also reported [60].

**Figure 2 molecules-27-07585-f002:**
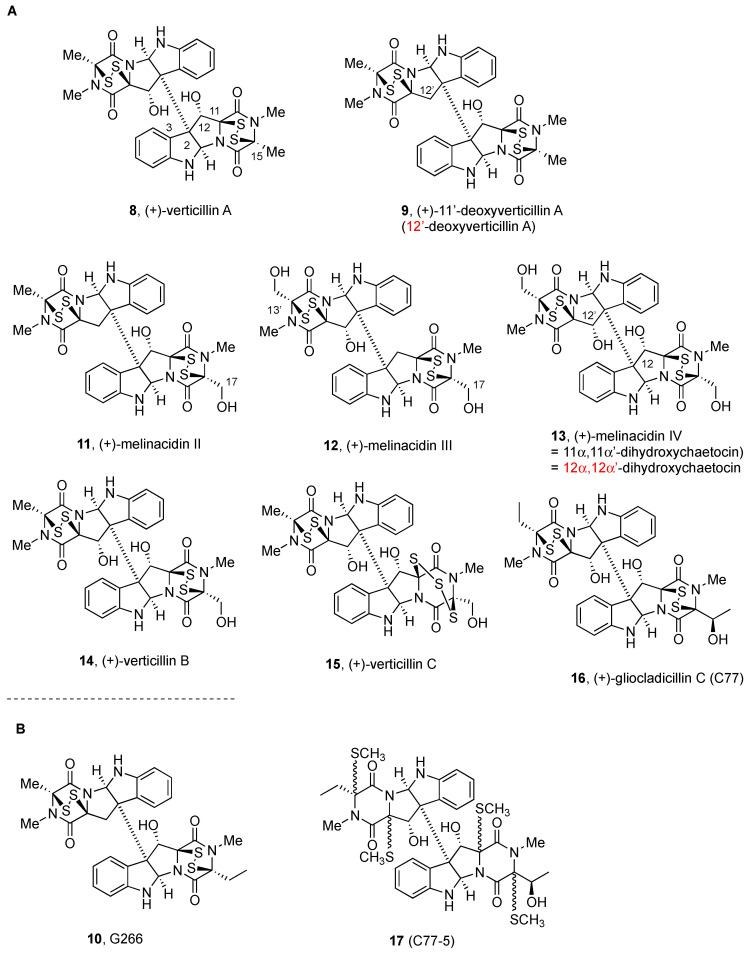
(**A**) Symmetrical and nonsymmetrical BPI ETP alkaloids derived from Trp, Ala, Ser, and Thr and oxidized and reduced derivatives; (**B**) Synthetically modified BPI ETPs.

(+)-Gliocladicillin C or C77 (**16**, Figure 2) was first isolated from the *Ophiocordyceps sinensis*-associated fungus *Clonostachys rogersoniana* [89]. Together with (+)-gliocladicillin A (**5**), (+)-gliocladicillin B (**6**, Figure 1), and 12’-deoxyverticillin A (**9**, Figure 2), (+)-gliocladicillin C (C77) (**16**) was more recently isolated from *O. sinensis*–associated strain of *Clonostachys rogersoniana* [78].

Biological evaluation of (+)-gliocladicillin C (C77) (**16**) and the synthetic disulfide-cleavage product termed C77-5 (**17**, Figure 2B) revealed that the former could induce caspase-dependent apoptosis and autophagy in human tumor cells (HeLa, T24, A549, HCT116, MCF7, and SW480), whereas derivative C77-5 (**17**) did not induce reactive oxygen species production and PARP cleavage [90]. Induction of phosphorylation of AMP-activated protein kinase and stimulation of autophagy was found to affect the glycolytic pathway. Therefore, it was concluded that the disulfide functionality in (+)-gliocladicillin C (C77) (**16**) was required for the induction of caspase-dependent apoptosis but not for the activation of autophagy.

(+)-Sch 52900 (**18**) and (+)-Sch 52901 (**19**, Figure 3) were first isolated, together with (+)-verticillin A (**8**, Figure 2) from the fermentation broth of the fungal culture *Gliocladium* sp. (SCF-1168) [91]. The identification relied on the structural similarities noted with previously described (+)-verticillin A (**8**) [20]. The nonsymmetrical nature of (+)-Sch 52900 (**18**) was concluded by the presence of apparently split signals in the ^13^C NMR spectrum. The identification of signals for the secondary alcohol in place of the methyl group of (+)-verticillin A (**8**) suggested the structure for (+)-Sch 52900 (**18**). (+)-Sch 52901 (**19**) was more easily identified as the nonsymmetrical structure having methyl and ethyl substituents at C3/C3’ of the dioxopiperazine units. The absolute configuration of these metabolites [91] was assigned by comparison of optical rotation data and circular dichroism spectral data with those of (+)-verticillin A (**8**) [20]. 

The three compounds prevented serum-stimulated transcription of the human c-fos promoter using a *fos/lac* Z reporter gene assay, with IC_50_ values of 1.5, 18, and 0.5 µM for (+)-Sch 52900 (**18**), (+)-Sch 52901 (**19**), and (+)-verticillin A (**8**, Figure 2), respectively [91]. The inhibition of both phorbol ester-induced *c-fos* induction and serum-induced JE induction without affecting RNA synthesis suggested that (+)-verticillin A (**8**) could be acting at a very early stage of the signaling pathway, leading to c-*fos* proto-oncogene induction, which is involved in cell proliferation [91].

After screening of fungal cultures for differentiation-inducing metabolites, (+)-Sch 52900 (**18**, Figure 3) was also identified in cultures of the imperfect fungus *Gliocladium*, strain 4–93, as the component strongly inducing differentiation and apoptosis of HL-60 cells. The effect was related to the induction of the cell cycle inhibitor protein p21 and the interference with the AP-1 signaling pathway [92].

(+)-Sch 52900 (**18**) and (+)-Sch 52901 (**19**, Figure 3) were also obtained from the organic extracts by the activity-guided fractionation of filamentous fungi of the Bionectriaceae strains MSX 64546 and MSX 59553. Potent cytotoxic effects were noted in assays against HT-29, H460, SF-268, MCF-7, and MDA-MB-435 cell lines. (+)-Sch 52900 (**18**) and (+)-verticillin A (**8**, Figure 2) inhibited the specific binding ability of activated p53 subunits of NF-κB in the nucleus of HeLa cells, with IC_50_ values of 0.5 and 0.1 µM, respectively [79].

**Figure 3 molecules-27-07585-f003:**
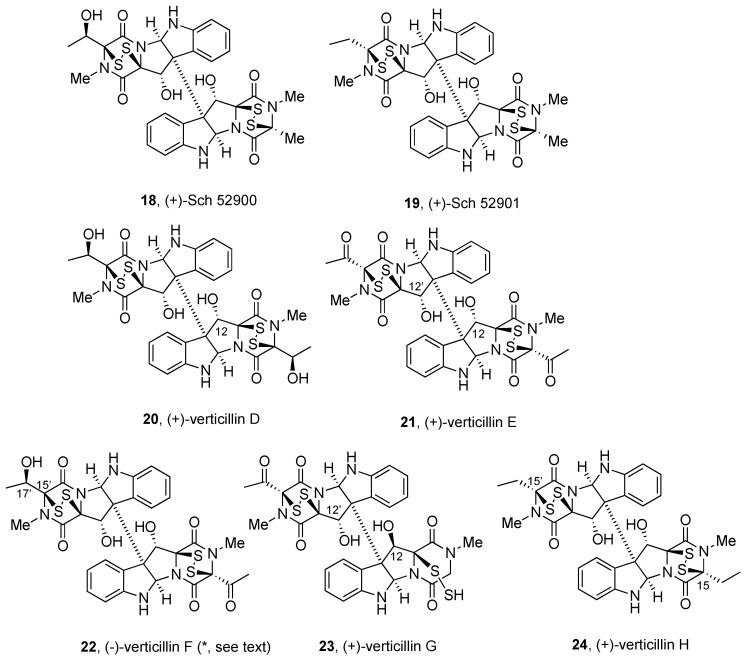
Symmetrical and nonsymmetrical BPI ETP alkaloids derived from Trp, Ala, Ser, and Thr and oxidized/reduced derivatives. *Absolute configuration uncertain.

Verticillins D–F (**20**–**22**, respectively, Figure 3) were isolated from solid-substrate fermentation cultures of the sclerotical mycoparasite *Gliocladium catenulatum* strain NRRL 22970 as a colonist of *Aspergillus flavus* Link:Fr. Sclerotia, which were buried for 2 years in a Georgia cornfield [87]. (+)-Verticillin D (**20**) was also obtained from an ethyl acetate extract of the endophytic fungus *Bionectria ochroleuca* collected from leaf tissues of the mangrove plant *Sonneratia caseolaris* from Hainan Island in China [93]. 

Identification of (+)-verticillin D (**20**) was based on the analysis of the NMR data of a triacetate derivative (retaining a hydroxyl group at C-12) generated by preferential formation upon treatment with acetic anhydride in pyridine at ambient temperature. Analysis of HMBC and HSQC data for the nonsymmetrical triacetate revealed the presence of three acetyl groups and a double set of signals for the remaining elements [87]. The NMR data for this symmetrical dimer were compared with that of (+)-verticillin A (**8**, Figure 2) [86] and found to differ by the nature of the amino acids, which are two threonine units for (+)-verticillin D (**20**). The conformationally constrained triacetyl derivative enabled the structural elucidation of (+)-verticillin D (**20**) through correlations of the acetyl groups with additional substituents [87].

Comparison of the spectroscopic data of (+)-verticillin E (**21**), with those of (+)-verticillin D (**20**), allowed for proposing similar structures, with the ketone of (+)-verticillin E (**21**) instead of the secondary alcohols in **20**. Similarly, two slightly different sets of signals of the nonsymmetrical structure were noted for (−)-verticillin F (**22**), with the presence of a quartet for an oxygenated methine proton at δ 4.63 (H-15’) coupled to a methyl doublet at δ 1.54 ppm (CH_3_-17’), and a methyl ketone (δ 2.50 ppm). 

Although the NMR analysis allowed for proposing for (−)-verticillin F (**22**) a structure resulting from the formal combination of each unit of (+)-verticillin D (**20**) and (+)-verticillin E (**21**) [87], the sign of the optical rotation for (−)-verticillin F (**22**), [α_D_] = −80 (*c* 0.1 mg/mL, MeOH), might indicate to be an enantiomer of the remaining alkaloid family members of the same class, including (+)-verticillin E (**20**). 

(+)-Verticillin E (**21**) and (−)-verticillin F (**22**) showed antibacterial activity at 100 µg/disk in standard Petri plate assays affording inhibitory zone sizes of 23 to 26 mm against *Bacillus subtilis* (ATCC 6051) and of 11 to 14 mm against *Staphylococcus aureus* (ATCC 14053) [87]. In cytotoxicity assays on the murine lymphoma L5178Y cell line, (+)-verticillin D (**20**) showed cytotoxic activities, with a 0.5% survival rate at 10 µg/mL and EC_50_ < 0.1 µg/mL (cf. EC_50_ = 6.40 µg/mL for kahalalide F used as positive control) [93].

(+)-Verticillin G (**23**, Figure 3) was isolated, together with (+)-verticillin D (**20**), from the mycelium of liquid fermentation cultures of the fungal strain *Bionectra byssicola* F120 [94]. The spectroscopic data were similar to those of (+)-verticillin D (**20**) [87], but in a nonsymmetrical structure, with major differences noted in HMBC data for the isolated methyl, ketone carbonyl, and isolated methylene on the DKP unit. These data revealed the lack of one bicyclic dithiodioxopiperazine and the presence of a hydrodisulfide functionality. Given the similar [α] value of (+)-verticillin G (**23**) and (+)-verticillin D (**20**), the relative and absolute configurations were proposed to be the same as that of the latter, with the exception of C-12 [94].

Antibacterial activities against *S. aureus* (*S. aureus* RN4220 and *S. aureus* 503), MRSA (*S. aureus* CCARM 3167 and *S. aureus* CCARM 3506) and QRSA (*S. aureus* CCARM 3505 and *S. aureus* CCARM 3519) were determined for both natural products. Whereas (+)-verticillin G (**23**) showed stronger activity on QRSA (MIC = 3 µg/mL) than against wild strains and MRSA (MIC = 10 µg/mL), (+)-verticillin D (**20**) exhibited stronger activity on wild strains and MRSA (MIC of 3 µg/mL) [94].

(+)-Verticillin H (**24**) was isolated following a bioactivity-directed fractionation of the organic extract of two filamentous fungi of Bionectriaceae, strains MSX 64546 and MSX 59553 from the Mycosynthetix library, together with Sch (+)-52900 (**18**), (+)-verticillin A (**8**, Figure 2), (+)-gliocladicillin C (**16**, Figure 2), (+)-Sch 52901 (**19**, Figure 3), (+)-12’-deoxyverticillin A (**9**, Figure 2), and (+)-gliocladicillin A (**5**, Figure 1) [79]. Comparison of the NMR data of (+)-verticillin H (**24**) with those of (+)-verticillin A (**8**, Figure 2) suggested the presence of an additional methylene unit. HMBC experiments allowed for establishing the connection of the ethyl side chain to the C-15 and C-15’ positions of each monomeric unit.

Similar to the other analogs isolated from the same source, (+)-verticillin H (**24**) displayed potent cytotoxic activity against HT-29, H460, SF-268, MCF-7, and MDA-MB-435 cell lines, with IC_50_ values of 0.04, 0.30, 0.33, 0.49, and 0.10 µM, respectively [79].

#### 2.2.2. Gliocladines and Chetracins 

(+)-Gliocladine A and (+)-gliocladine B (**25** and **26**, respectively, Figure 4), together with C3-β-indolyl-substituted gliocladines C, D, and E (not shown), were isolated from wheat solid-substrate fermentation of *Gliocladium roseum* 1A [80], along with (+)-verticillin A (**8**, Figure 2), (+)-12’-deoxyverticillin A (**9**, Figure 2), (+)-Sch 52900 (**18**, Figure 3), and (+)-Sch 52901 (**19**, Figure 3) [91].

**Figure 4 molecules-27-07585-f004:**
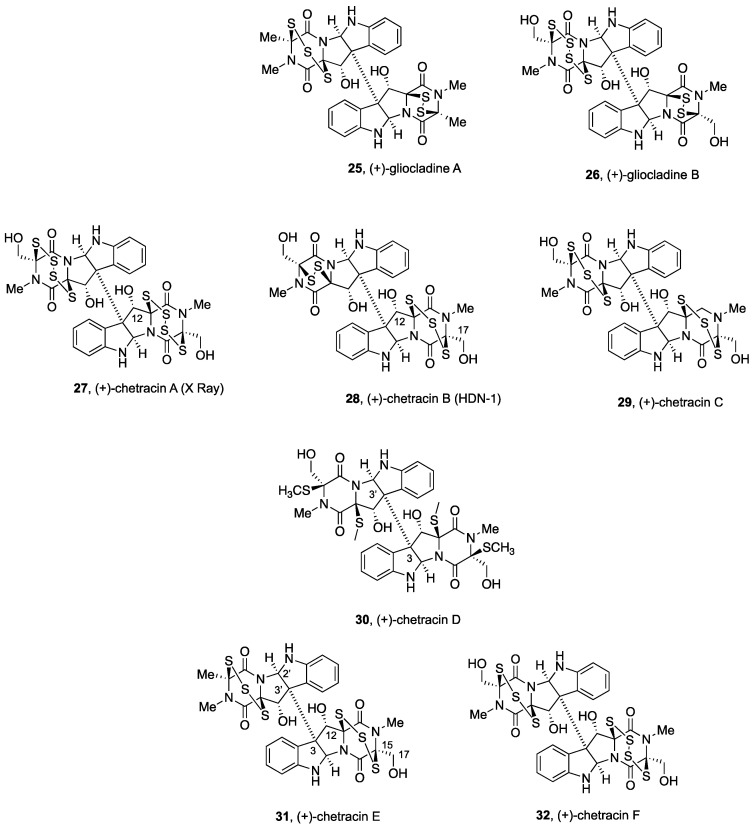
Nonsymmetrical BPI ETPs alkaloids derived from Trp, Ser, and Ala.

Comparison of their NMR data with those described for the already-identified analogues (**8**, **9**, **18**, and **19**) [76,91] allowed the elucidation of their gross structures. (+)-Gliocladine A (**25**) contains one more sulfur atom than (+)-verticillin A (**8**) [91], and both share one DKP subunit with a disulfide bridge. Although structurally related to (+)-gliocladine A (**25**), for the nonsymmetrical structure of (+)-gliocladine B (**26**), the FABMS analysis allowed for identifying a tetrasulfide in one of the monomeric units, with hydroxymethyl substituents at the dithio-DKP subunit [80].

Potent nematicidal effects of these compounds against *C. elegans* and *P. redivivus* were measured (ED_50_ in µg/mL of 25 and 50 for (+)-gliocladine A (**25**), 30 and 25 for (+)-gliocladine B (**26**), 30 and 80 for (+)-verticillin A (**8**), 10 and 40 for (+)-12’-deoxyverticillin A (**9**), 50 and 80 for (+)-Sch 52900 (**18**), and 50 and 50 for (+)-Sch 52901 (**19**)), indicating that the number of sulfur atoms in the subunits is not relevant for the nematicidal activity of these natural products [80].

(+)-Chetracin A (**27**) was first isolated from *Chaetomium* sp. in the course of studies on mycotoxin production by the fungi of the species *Chaetomium abuense* Lodha and *C. retardatum* Carter and Khan, which also generated (+)-12α,12α’-dihydroxychaetocin or melinacidin IV (**13**, Figure 2) [62,75].

Isolation of (+)-chetracin A (**27**) was carried out after transformation of the natural product into the corresponding acetate (Ac_2_O, pyridine), purification by HPLC, and saponification. Analysis of the spectroscopic data suggested (+)-chetracin A (**25**) to be a tetrathio homologue of 12α,12α’-dihydroxychaetocin or melinacidin IV (**13**, Figure 2). To confirm their structural analogy, (+)-chetracin A (**27**) was treated with PPh_3_-NH_4_OH-MeOH to generate (+)-12α,12α’-dihydroxychaetocin (**13**). The absolute configuration was assigned based on the similar CD spectra of the acetates derived from both natural products [62,75].

X-ray diffraction analysis using the CuKα radiation monochromated by a graphite plate confirmed the structure and the absolute configuration of synthetic (+)-chetracin triacetate (not shown) using the anomalous dispersion method [62,75]. It was noticed that the dioxopiperazine rings become stabilized in the boat conformation, which appears to be more planar than in the case of structural analogs with disulfide bridges. Interestingly, the four terminal S–S bonds (with bond distances of 1.996, 2.005, 2.002, and 2.011 Å) were found to be shorter than the central S–S bonds (bond distances of 2.066 and 2.0192 Å).

(+)-Chetracin B or HDN-1 (**28**) [68] and (+)-chetracin C (**29**), together with the tetrakis(methylsulfanyl) derivative named (+)-chetracin D (**30**, Figure 4), were isolated from the Antarctic fungus *Oidiodendron truncatum* GW3-13 obtained from soil under lichens near the Great Wall station (Chinese Antarctic station). Twinned resonances were noted in the ^1^H NMR spectra, which was indicative of the dimeric nature, and slightly different monomers were proposed for the relative composition of the two units of (+)-chetracin B (**28**) [95]. The structural similarities to dimeric (+)-12α,12α’-dihydroxychaetocin or melinacidin IV (**13**, Figure 2) allowed for suggesting for (+)-chetracin B (**28**) the presence of a trisulfide bond at the second monomeric subunit on the basis of the downfield chemical shift of C-12 due to the deshielding effect observed for other members of the series [88]. The relative configuration was proposed after analysis of the NOESY spectrum, and was further confirmed after conversion to the tetrakis(methylsulfanyl) derivative upon treatment with NaBH_4_ and MeI, namely, (+)-chetracin D (**30**), a natural product also isolated from the same strain [95]. The absolute configuration could not be determined unambiguously by comparison of its CD spectrum with that of (+)-12α,12α’-dihydroxychaetocin or melanicidin IV (**13**, Figure 2) [73], since it lacked the characteristic absorption at about 270 nm and the positive Cotton effect at 230 nm. However, when a solution of (+)-chetracin B (**28**) in DMSO was allowed to stand for about 2 weeks at ambient temperature, a mixture of (+)-chetracin C (**29**) and melanicidin IV (**13**, Figure 2) was obtained [95], likely through disproportionation of polysulfides via a free radical mechanism [96].

Symmetrical dimer (+)-chetracin C (**29**) also contains one more sulfur atom on each unit than (+)-12α,12α’-dihydroxychaetocin or melinacidin IV (**13**). Given the similar chemical shifts in their ^1^H and ^13^C NMR spectra exhibited by both compounds, they were proposed to differ by the presence of trisulfide bridges as structural components in (+)-chetracin C (**29**). The absolute configuration was assumed to be the same of melanicidin IV (**13**, Figure 2), considering their putative common biogenetic origin [95].

(+)-Chetracin B or HDN-1 (**28**) [68,95] was also isolated, together with (+)-12α,12α’-dihydroxychaetocin or melinacidin IV (**13**, Figure 2) [73], from the algicolous fungus *Westerdykella reniformis* obtained from Prince Edward Island, Canada [85], and identified by comparison of the ^1^H NMR data with those previously described [95]. 

Samples of (+)-chetracin B (**28**) and (+)-12α,12α’-dihydroxychaetocin or melanicidin IV (**13**, Figure 2) isolated from the algicolous fungus *Westerdykella reniformis* were subjected to antimicrobial testing against the drug-resistant Gram-positive bacteria methicillin-resistant *Staphylococcus aureus* (MRSA) and vancomycin-resistant *Enterococcus faecium* (VRE) and the Gram-negative bacterium *Proteus vulgaris*. Both compounds showed MIC values (0.7 and 0.7 µM, respectively) and half-maximal inhibitory concentrations IC_50_ (0.1 and 0.2 µM, respectively) slightly lower than those of vancomycin (MIC 1.4 and IC_50_ 0.6 µM) against MRSA. However, they were less efficacious than rifampicin and ciprofloxacin against VRE and *P. vulgaris*, respectively [85]. 

(+)-Chetracin C (**29**) was shown to act as Hsp90α inhibitor binding to the C-terminus and reducing the levels and active forms of oncoproteins EGFR, Stat3, Akt, and Erk. Similar to (+)-chaetocin A (**1**), it was reported to act as an inhibitor of Hsp90 and to regulate SUV39H1 histone lysine methyltransferase stability [68].

Likewise, (+)-chetracin D (**30**) showed similar NMR data as (+)-12α,12α’-dihydroxychaetocin or (+)-melinacidin IV (**13**, Figure 2), but contained four *S*-methyl groups. ^1^H NMR analysis of solutions of **30** in DMSO-d_6_ indicated a 1:3.3 mixture of rotational isomers about the central C-3—C-3’ bond for the natural product. NOESY data confirmed the same relative configuration as (+)-chetracin B (**28**), and the absolute configuration was also assigned considering that it is formally a reduction/methylation product of **28 [95]**.

(+)-Chetracin B (**28**) and (+)-12α,12α’-dihydroxychaetocin or (+)-melinacidin IV (**13**, Figure 2) exhibited potent cytotoxic activity against five human cancer cell lines (HCT-8, Bel-7402, BGC-823, A549, and A2780) in the nanomolar range (from 0.003 to 0.054 µM), with minor differences among them, which appears to indicate that the number of sulfur atoms in the series is less important for their activities [95].

(+)-Chetracin E (**31**) and (+)-chetracin F (**32**), together with (+)-chetracin C (**29**), were isolated from a marine strain of the fungus *Acrostalagmus luteoalbus* HDN13-530 obtained from soil in Liaodong Bay [97]. The ^1^H NMR of (+)-chetracin E (**31**) was nearly superimposable to that of (+)-chetracin C (**29**) with one hydroxymethyl of the latter being replaced by a methyl group in **31**, as confirmed by HMBC correlations. The relative configuration was established by NOESY experiments [97]. Moreover, NOESY correlations after generation of the tetrakis(methylsulfanyl) derivative (upon treatment with NaBH_4_ and MeI) indicated H-12 and H-12’ to be *cis* to the sulfur bridge [97]. The same absolute configuration of (+)-chetracin D (**30**) was assigned to (+)-chetracin E (**31**) based on similar CD spectra and almost identical CD curves [97].

(+)-Chetracin F (**32**) showed an additional sulfur atom when compared with (+)-chetracin E (**31**), and therefore, a tetrasulfide bridge was proposed as part of the DKP ring in one of the monomeric units [97]. The same absolute configuration of (+)-chetracin E (**31**) was also confirmed by the formation of an identical tetrakis(methylsulfanyl) derivative. In addition, when solutions of (+)-chetracin F (**32**) in DMSO were left at room temperature for 2 weeks, partial conversion to (+)-chetracin C (**29**) was noted, in a process that is likely induced by radical reactions, which also confirmed the similar absolute configuration for both natural products [97].

Biological evaluation using the MTT assay showed strong cytotoxic activities for (+)-chetracin E (**31**), (+)-chetracin F (**32**), and (+)-chetracin C (**29**), with similar potency than (+)-chetracin B (**28**) [95], against five cancer cell lines (A549, HCT116, K562, H1975, and HL-60) at low µM to nM concentrations with IC_50_ values between 0.2 and 3.6 µM (cf. doxorubicin hydrochloride showed IC_50_ values from 0.2 to 0.8 µM, but 0.02 µM for HeLa cells). (+)-Chetracin E (**31**) was found to be the most potent of the series on H1975 cells (IC_50_ 0.2 µM) [97]. 

Modeling studies suggested binding of (+)-chetracin E (**31**), (+)-chetracin F (**32**), and (+)-chetracin C (**29**) to the C-terminal Hsp90 client oncoprotein, which is in agreement with the reduction of the expression of EGFR and Akt and the active forms of EGFR, Stat3, Akt, and Erk in H1975 cells upon treatment with these natural products [97].

#### 2.2.3. Leptosins and Preussiadins 

The leptosins were mainly isolated by Numata and coworkers, between 1994 and 2005, from a strain of the fungus *Leptosphaeria* sp., obtained from the surface of the brown alga *Sargassum tortile* collected from Tanabe Bay, Japan [57,88,98,99]. It includes now 21 members. At least one Val residue is present in the group of natural products, but a varying number of sulfur atoms can be present in the sulfur-rich bridge [13].

Dimeric leptosins have been divided into four groups: (a) BPI-ETP dimers with polysulfide bonds and the same relative configuration, (b) BPI-ETP dimers with polysulfide bonds and opposite relative configuration, and (c) BPI-ETP dimers with one polysulfide and the same relative configuration; (d) BPI-ETP dimers with sulfide and disulfide units, formally with broken thio-bridge structures.

**Figure 5 molecules-27-07585-f005:**
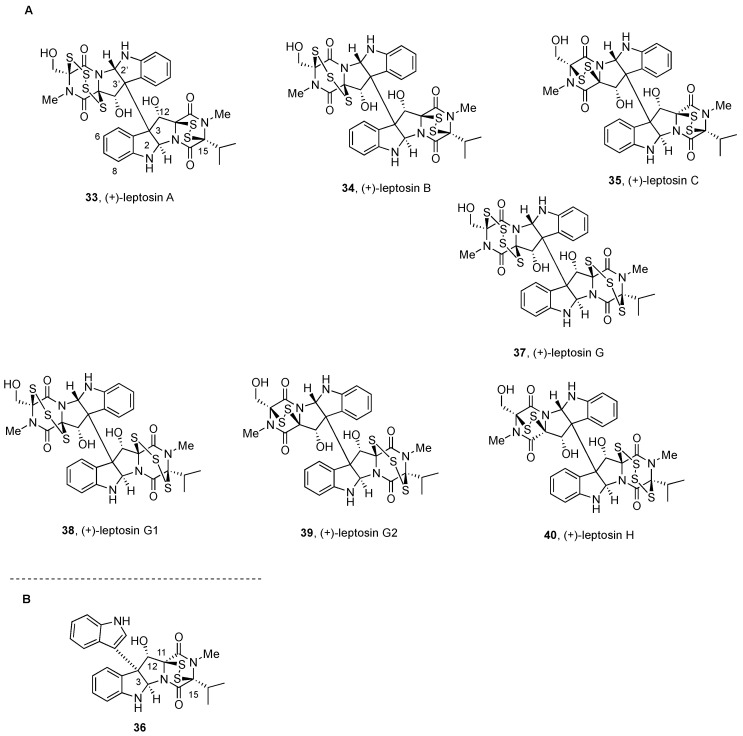
(**A**) Nonsymmetrical BPI-ETP alkaloids derived from Trp, Ser, and Val (I). (**B**) Partial degradation derivative of **33** and **34**.

(+)-Leptosin A, (+)-leptosin B, and (+)-leptosin C (**33**, **34**, and **35**, respectively) were isolated, together with the nondimeric indole-DKP analogs leptosins D–F (not shown), from the mycelium of a strain of *Leptosphaeria* sp. attached to the marine alga *Sargassum tortile* [57].

The transformation of (+)-leptosin A (**33**) into the methyl derivatives upon reduction with NaBH_4_ and treatment with MeI in pyridine allowed for locating the tetrasulfide and disulfide bridges in the hydroxymethyl- and isopropyl-bearing dioxopiperazines, respectively [57]. The planar structure could then be determined based on HMBC correlations. The relative configuration was deduced from NOE enhancement spectral analysis of both (+)-leptosin A (**33**) and the *S-S* reduced/methylated derivative. The relative configuration at C-2’ and C-3’ determined for (+)-leptosin A (**33**) [57] was found to differ from those of structurally related (+)-chaetocin A (**1**, Figure 1) [19,59], and it became the first member of this family with the opposite relative configurations at those carbon atoms.

Similar identification protocols for (+)-leptosin B and (+)-leptosin C (**34** and **35**, respectively) indicated IR, UV, and NMR spectra closely related to those of parent (+)-leptosin A (**33**), and differences were noted for some ^13^C NMR signals. FABMS analysis of the natural products and the methyl sulfide derivatives obtained by reduction/methylation demonstrated the presence of hydroxymethyl-bearing dioxopiperazine rings on the structures of (+)-leptosin B and (+)-leptosin C (**34** and **35**), with three and two sulfur bridges, respectively [57].

The negative band at 271 nm in the CD spectrum of (+)-leptosin C (**35**) due to the S→CO charge transfer transition when compared in strength with that of a di-*O*-acetyl derivative of (+)-chaetocin A (**1**, Figure 1) suggested the same configuration (*S*), despite the fact that they differ in the configuration of one of the DKPs, which was corroborated by NOE experiments, and the absolute stereostructure of the series (**33–35**) was assigned accordingly [57].

(+)-Leptosin A (**33**) and (+)-leptosin C (**35**) were more recently obtained, together with (+)-preussiadin A (**45**) and (+)-preussiadin B (**46**, Figure 6), from the mycelia of a *Preussia typharum* isolate that originated from a bottomlandforest-derived soil sample [100].

The ^1^H NMR spectrum of (+)-leptosin C (**35**) in CDCl_3_ solution showed broad proton resonances [100], which was suggested to be likely due to the presence of conformers [57]. VT NMR experiments (down to −60 °C) allowed for observing two distinct sets of sharpened proton signals. However, in acetone-d_6_, a single stereoisomer was stabilized at low temperature [100], a behavior that was not observed for (+)-leptosin A (**33**) and other DKPs [57].

The absolute configuration was assigned following conversion of (+)-leptosin B (**34**) into a mixture of the 3-substituted indole derivative **36** (a natural product termed leptosin D, Figure 5B) and (+)-leptosin C (**35**) upon treatment with triphenylphosphine at 60 °C. In addition, the ring-contracted analogue with a disulfide connector in leptosin C (**35**, Figure 5) was obtained from those of (+)-leptosin A (**33**, Figure 5) or (+)-leptosin B (**34**, Figure 5) under the same conditions but at ambient temperature [57]. Bis(methylsulfanyl) and tetrakis(methylsulfanyl) derivatives of leptosin A were also obtained and fully characterized upon treatment of the natural product with sodium borohydride and methyl iodide [57].

These compounds showed potent cytotoxicity against cultured P388 cells with values of ED_50_ = 1.85 × 10^−3^ µg/cm^−3^ for (+)-leptosin A (**33**), ED_50_ = 2.40 × 10^−3^ µg/cm^−3^ for (+)-leptosin B (**34**), and ED_50_ = 1.75 × 10^−3^ µg/cm^−3^ for (+)-leptosin C (**35**), and also significant antitumor activity against sarcoma 180 ascites [57]. (+)-Leptosin A and (+)-leptosin C (**33** and **35**, respectively) were also shown to display low nanomolar potency towards the human pancreatic MIA PaCa-2 cancer cell line [100]. (+)-Leptosin A (**33**) was proposed to induce apoptosis in human lymphoblastoid RPMI8402 and embryonic kidney cells through the inhibition of topoisomerase I and the Akt/protein kinase B survival pathway [101].

Leptosins G, G_1_, G_2_, and H (**37**, **38**, **39**, and **40**, respectively) were isolated, together with leptosins A–C (**33–35**), from the mycelium of a strain of *Leptosphaeria* sp. stuck on the marine alga *Sargassum tortile* [99]. The molecular formula and the general features of the UV, IR, and NMR data suggested leptosins G, G_1_, G_2_, and H (**37**, **38**, **39**, and **40**, respectively) to be sulfur analogs of leptosins A–C (**33**–**35**).

MS and spectroscopic data for (+)-leptosin G (**37**) were consistent with the presence of a tetrasulfide bridge at the hydroxymethyl-bearing structural component, which was also in agreement with the easier loss of the [MH - 4S]^+^ peak in the region bearing a hydroxymethyl group, and then a trisulfide bridge was ascribed to the moiety bearing an isopropyl substituent [99]. The same MS analysis suggested the presence of two trisulfide bridges in (+)-leptosin G1 (**38**). In the case of (+)-leptosin G_2_ (**39**), transformation of the natural product into the bis(methylthio)- and tetrakis(methylthio)-derivatives as indicated above allowed for establishing its stereostructure. (+)-Leptosin H (**40**) had the same molecular formula as (+)-leptosin G_1_ (**38**), but with the disulfide and the tetrasulfide bridges in opposite DKP units when compared with (+)-leptosin A (**33**).

Cytotoxic activities were examined in a P388 lymphocytic leukemia test system in cell culture. The series of compounds **37–40** exhibited cytotoxic levels similar to those of the analogs leptosins A–C (**33–35**), which indicated that the location and number of sulfur atoms on the DKP units had little influence on the reported activities [99].

(+)-Leptosin K, (+)-leptosin K_1_, and (+)-leptosin K_2_ (**41**–**43**, respectively, Figure 6) were later isolated from the same fungal strain [88], and determined to contain Val residues on the skeletons of both monomeric epipolysulfanyldioxopiperazine units.

The spectroscopic analysis of (+)-leptosin K (**41**) and the bis(methylsulfanyl) and tetrakis(methylsulfanyl) derivatives obtained upon treatment with NaBH_4_ and MeI allowed for establishing the planar structure of the natural product, most notably the presence of the disulfide bridge in the isopropyl-bearing dioxopiperazine moieties. Interpretation of the NOE data for the natural product and the tetrakis(methylsulfanyl) derivative led to proposing the relative configuration of the disulfide bridges and the chair conformations of the dioxopiperazines with the isopropyl group in an axial arrangement. In addition, the NOE observed between H-2 and H-2’ allowed the assignment of the relative configuration of these compounds. The relative configuration of C-15’ and C-11’ in (+)-leptosin K (**41**) was therefore the same reported for leptosins A–C (**33–35**) [57] and also for leptosins G, G1, G2, and H (**37**–**40**, Figure 5) [99].

The X-ray analysis of (+)-leptosin K (**41**) confirmed the relative configuration of the series. Interestingly, it showed the presence of two closely resembling pairs of conformers in a single crystal, which differ by the rotation angle of the monomeric units around the C3—C-3’ bond. Rotational angles of the C3—C-3’ and C3’—C-12’ bonds were −166°, −165°, 74°, and 70°. Conformational studies in solution indicated the presence of mixtures of two conformers in CDCl_3_ (Figure 6B) which became single conformers in pyridine-d_5_. The comprehensive NMR study allowed for establishing that a combination of chemical shift values for aryl protons H-5 and H-5’ and NOE correlations between H-2, H-5, H5’, and H-12’ could be used to assign the conformation of the natural products in solution as a function of the solvent [88].

(+)-Leptosin K_1_ (**42**) was also shown to exist as a 3:2 mixture of conformers in CDCl_3_ solution at 0 °C and as a single conformer in pyridine-*d_5_*. Similar to the other congeners, treatment with NaBH_4_ and MeI and isolation of the thiomethyl derivatives led to proposing the heterodimeric structure with disulfide and trisulfide bridges in the two units, through analysis of FABMS fragments [88].

From leptosin K2 (**43**, Figure 6A) treatment with PPh_3_ at ambient temperature afforded leptosin K1 (**42**, Figure 6A). Heating the latter with the same reagent to 60 °C afforded a mixture of leptosin K (**41**, Figure 6A) and the C3-indole derivative **44** (Figure 6C) [88].

The conformational equilibrium was likewise shown in the complex NMR spectra of (+)-leptosin K_2_ (**43**) in CDCl_3_, and further confirmed by NOESY data [88]. Comparison of NMR spectra with those of parent (+)-leptosin K (**41**) led to recognizing a monomeric unit with the same disulfide structure, and another one with a tetrasulfide bridge, as supported by analysis of FABMS fragments. NOESY cross peaks were similar to those of parent (+)-leptosin K (**41**), suggesting the presence of a 1:1 mixture of conformers in CDCl_3_ and a single isomer in pyridine-*d_5_*.

Finally, the absolute configuration of the series was assigned following conversion of (+)-leptosin K_1_ (**42**) and (+)-leptosin D (**36**, Figure 5A) into the C3-substituted indole derivatives (**44** and **36**, respectively) upon treatment with triphenylphosphine. These compounds are diastereomers and share the configuration at C-12/C-12’. Analysis of CD spectra and NOE data led to proposing the *S* configuration at both C-11 and C-15 of **44** and the *R* configuration at the same atoms in **36**. Given the large upfield shift (δ 0.41 ppm) of the signal at H-5’ in **44** with respect to that of **41**, the relative configuration of the hydroxyl group at C-12 appears to exert a favorable influence upon rotation about the C3-C3’ bond in these alkaloids. Finally, an opposite sign of the charge transfer band at ca. 266 nm in the monodisulfide derivative led to assigning the *R* configuration at C-3’ and C-12’ for leptosin K_1_ (**42**) and related natural products leptosin K_2_ (**43**) and leptosin K (**41**) [88].

Potent cytotoxic activities (ED_50_ in µg/mL of 3.8 × 10^−3^ for (+)-leptosin K, 2.2 × 10^−3^ (+)-leptosin K_1_, and 2.1 × 10^−3^ for (+)-leptosin K_2_) were found for these natural products (Figure 6) when tested in the P388 lymphocytic leukemia in cell culture [88]. 

**Figure 6 molecules-27-07585-f006:**
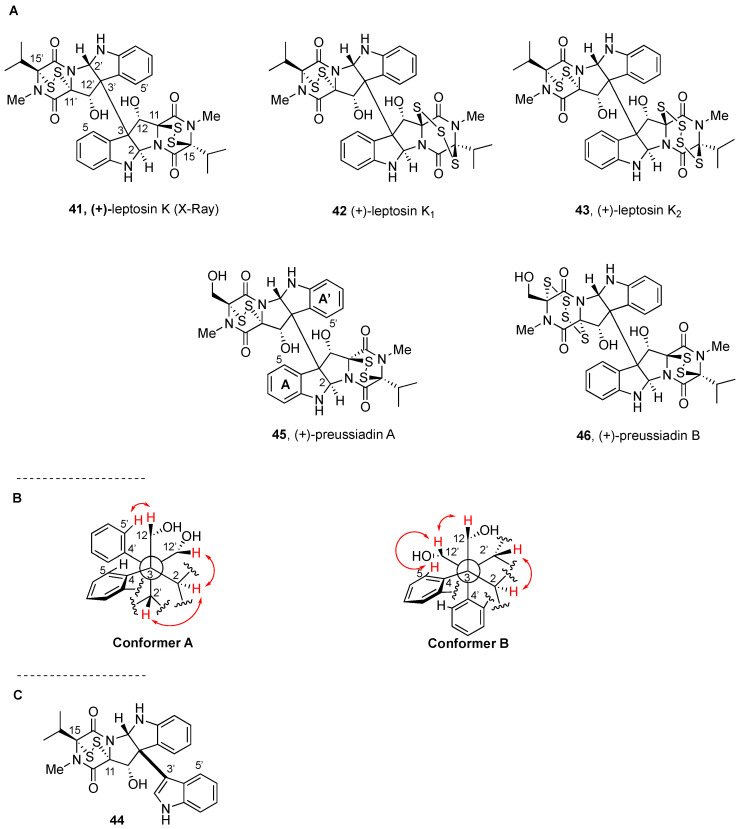
(**A**) Nonsymmetrical BPI-ETP alkaloids derived from Trp, Ser, and Val (II); (**B**) Conformers A and B (with the reported NOE correlations) of (+)-leptosin K (**41**); (**C**) Partial degradation derivative of (+)-leptosin K_1_ (**42**).

(+)-Preussiadins A (**45**) and B (**46**, Figure 6) were isolated together with (+)-leptosin C (**35**) and (+)-leptosin A (**33**, Figure 5) from the mycelia of a *Preussia typharum* isolate that originated from a bottomlandforest-derived soil sample [100]. Analysis of spectroscopic data led to proposing that (+)-preussiadin A (**45**) was a diastereomer of (+)-leptosin C (**35**, Figure 5) [57]. Although the ^13^C NMR data for the two compounds were nearly superimposable for the lower monomeric unit, significant variations were observed (with Δδ values greater than 2 ppm) among the chemical shifts assigned to the dithiodioxopiperazine ring of the upper monomeric fragment. This was ascribed to a change in the absolute configuration of the DKP ring. Disulfide cleavage and methylation as described above afforded the expected product and two minor *N*-monomethylated analogues at each of the formal dihydroindole nitrogen atoms. ROESY correlations in the *S*-methylated reduced derivative led to confirming that the configurations at C-11’ and C-15’ were inverted, thus identifying (+)-preussiadin A (**45**) as a diastereomer of (+)-leptosin C (**35**, Figure 5) [100].

The assignment of the absolute configuration of these compounds was based on a conformational search using Spartan’10 at the molecular mechanics level (MMFF) and TD-DFT computations (B3LYP with the 6-31+G* basis set) [102], including calculation of the VCD and ECD spectra of the six lowest-energy conformations and correlation with the experimental value [100].

Elemental analysis of (+)-preussiadin B (**46**) indicated the presence of two additional sulfur atoms. Differences on NMR spectroscopic data when compared with those of (+)- preussiadin A (**45**) led to proposing an upper unit with a tetrasulfide bridge between C-11’ and C-15’ for (+)-preussiadin B (**46**), while sharing both compounds the same absolute configuration [100].

As indicated above, ^1^H NMR spectra in CDCl_3_ at 25 °C confirmed the broad proton resonances for (+)-leptosin C (**35**, Figure 5), which were assigned to the interconversion of conformers as confirmed using ROESY experiments under variable low temperature conditions (−20, −40, and −60 °C). In contrast, a single isomer was observed when using acetone-*d_6_* as solvent. However, solutions of (+)-preussiadin A (**45**) and (+)-preussiadin B (**46**) in CDCl_3_ under the same conditions revealed just a single stereoisomer. ROESY experiments allowed the assignment of the *P* configuration to (+)-preussiadin A (**45**), and the *M* configuration to (+)-preussiadin B (**46**). Analysis of the ^1^H NMR spectrum of (+)-preussiadin A (**45**) revealed H-5 to be shifted upfield (to δ_H_ = 5.71 ppm) relative to H-5’ (δ_H_ = 7.70 ppm) due to the shielding effect of phenyl ring A’. In contrast, H-5’ for (+)-preussiadin B (**46**) was observed at an upfield position (to δ_H_ = 5.85 ppm) due to the shielding effect of the phenyl group A (Figure 6A) [100].

Potent cytotoxic activities against the human pancreatic cancer cell line MIA PaCa-2 were reported for (+)-preussiadin A (**45**) and (+)-preussiadin B (**46**), with IC_50_ values of 6.6 and 9.1 nM, respectively. (+)-Leptosin A and (+)-leptosin B (**33** and **34**, respectively, Figure 5) were less potent (33.6 and 43.1 nM, respectively) in the same assay. (+)-Preussiadin A (**45**) exhibited potent antiproliferative activity against each of the 60 cell lines of NCI-60, with an average GI_50_ = 14.8 nM, and cytotoxic effects against all solid tumor cell lines, with an average LG_50_ = 251 nM, but no cytotoxicity was noticed against the six leukemia cell lines tested [100]. It was also shown that (+)-preussiadin A (**45**) was able to overcome Pgp-mediated drug resistance. Although these cellular effects were determined to be persistent, short exposure times were sufficient to induce cytotoxicity. Neither mTORC1 inhibition nor effects on the levels of trimethylated H3K9 in HeLa cells were observed upon treatment with this compound. Furthermore, in vivo antitumor activity in a UACC-62 xenograft melanoma mouse model was noticed, although (+)-preussiadin A (**45**) has a very narrow therapeutic window [100].

(+)-Leptosin I and (+)-leptosin J (**47** and **48**, respectively, Figure 7), which share a tetrasulfide bridge, were also isolated from the mycelium of a strain of *Leptosphaeria* sp. OUPS-4 attached to the marine alga *Sargassum tortile* C. Agaroh (Sargassaceae) [98].

For (+)-leptosin I (**47**), a tetrasulfide bridge was proposed to span from C-11 to C-15, whereas in the second unit, no sulfur atoms were found. Instead, C-11’ was substituted with an oxygen atom. Analogs with *S*-Me groups were obtained upon reduction with NaBH_4_ and alkylation with MeI. Acetylation with Ac_2_O/pyridine afforded a diacetate showing the presence of a primary hydroxymethyl group at C-15’, a secondary hydroxyl group at C-12’, and a connection of C-12 with C-11´ by an ether linkage [98].

**Figure 7 molecules-27-07585-f007:**
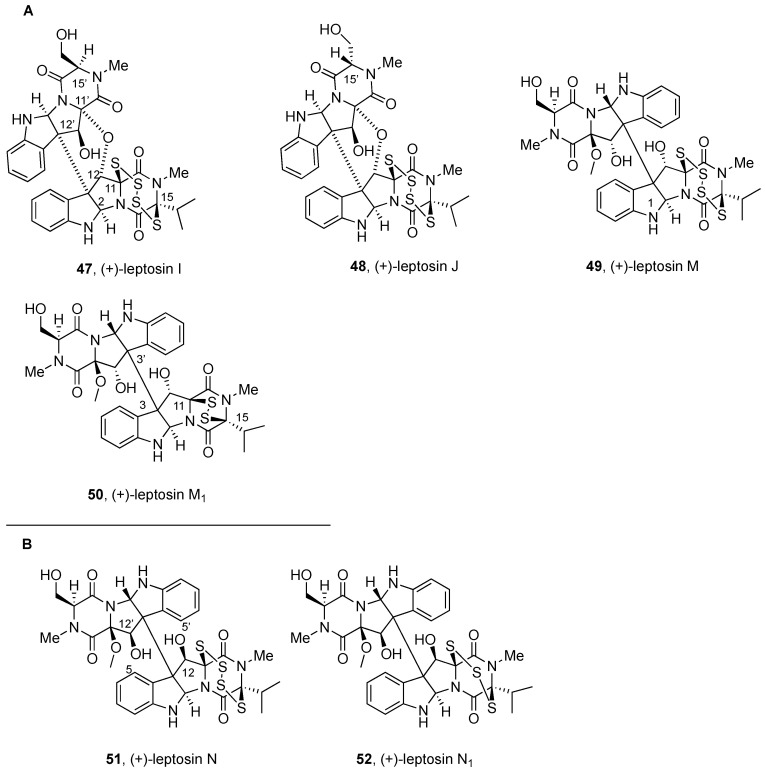
(**A**) Nondimeric BPI-ETP alkaloids with a polysulfide unit; (**B**) Nondimeric BPI-ETP alkaloids with a polysulfide unit and *S* configuration at C12/C12’.

Spectroscopic data for (+)-leptosin J (**48**, Figure 7) were very similar to those of (+)-leptosin I (**47**), with relevant chemical shift differences noted in the ^1^H NMR spectra for H-15’, the N-Me and CH_2_OH groups, and C-15’ and C-16’. However, similar cross peaks were observed in the NOESY spectra of these natural products. The structure of a stereoisomer of (+)-leptosin I (**47**) at C-15’ was proposed for (+)-leptosin J (**48**). The chemical shift differences were therefore ascribed to the axial or equatorial location of the hydroxymethyl on the dioxopiperazine rings, and their values led to suggesting that H-15’ displays axial and equatorial orientations in the structures of (+)-leptosin I (**47**) and of (+)-leptosin J (**48**), respectively. NOE enhancements confirmed the stereochemical arrangement of the substituents on the chair conformation of this subunit [98].

Significant cytotoxicity (ED_50_ of 1.13 and 1.25 µg/mL, respectively) was determined for (+)-leptosins I and (+)-leptosin J (**47** and **48**, respectively) in cell cultures of P388 lymphocytic leukemia [98].

(+)-Leptosin M, (+)-leptosin M_1_, (+)-leptosin N, and (+)-leptosin N_1_ (**49**–**52**, respectively, Figure 7) were later isolated from a strain of *Leptosphaeria* sp. OUPS-4 originally separated from the marine alga *Sargassum tortile* [103].

NMR analysis of (+)-leptosin M (**49**) allowed for proposing the planar structure, including the connection of the fragments and the unusual presence of the methoxy substituents at C11’, through analysis of the ^1^H-^1^H COSY and HMBC correlations. A tetrasulfide bridge in the isopropyl-substituted DKP was inferred to be present upon spectroscopic analysis of the methylsulfanyl derivative obtained after disulfide reduction with NaBH_4_ in the presence of CH_3_I. Positioning of the sulfur atoms was based on HMBC correlations, which indicated also the presence of the methyl group at N-1 due to undesired methylation [103]. The relative configuration was suggested based on NOE analysis of these derivatives in acetone-d_6_. Marfey’s analysis applied to the *N*-methylserine fragment derived from acid hydrolysis of (+)-leptosin M (**49**) revealed the L-(*S*)-configuration at that stereocenter and confirmed the stereostructure of the natural product.

(+)-Leptosin M (**49**) in acetone-d_6_ also exists as a B-type conformer (Figure 6B) [103], in contrast with (+)-leptosin K (**41**, Figure 6A), which adopts a single conformation of the A type (Figure 6B) [88], and (+)-leptosin K_1_ (**42**) and (+)-leptosin K_2_ (**43**, Figure 6A), which exist as a mixture of two conformers of A and B types about the C-3 and C-3’ bond. 

(+)-Leptosin M_1_ (**50**, Figure 7A) was confirmed to contain two less sulfur atoms than (+)-leptosin M (**49**) as deduced from the signals in the ^13^C NMR spectrum assigned to the dithiodioxopiperazine ring, and proposed to likewise adopt a B-type conformation [103]. 

(+)-Leptosin N (**51**, Figure 7B) shares the general formula of (+)-leptosin M (**49**). Following analysis of the differences in their NMR spectra, the signals for H-5 and H-5’ in (+)-leptosin N (**51**) appearing at lower (δ = 7.87 ppm) and higher (δ = 5.52 ppm) values relative to those of (+)-leptosin M (**49**) led to suggesting the structure of a stereoisomer at C-12 and C-12’ and the adoption of a B-type conformation in C_6_D_6_. The presence of a tetrasulfide bridge on the natural product was deduced from NMR analysis and conversion on its bis(methylthiosulfanyl) derivative, which showed two *S*-Me but no additional N-Me groups. HMBC analysis also suggested the presence of a tetrasulfide bridge, which was corroborated by NOE correlations. Marfey’s analysis revealed the L-(*S*)-configuration for the *N*-methylserine moiety [103]. 

(+)-Leptosin N_1_ (**52**, Figure 7) contains one less sulfur atom, and its structure with a trisulfide connection on the DKP unit was proposed after analysis of the differences in ^13^C NMR data with those of the isopropyl-bearing dioxopiperazine (+)-leptosin N (**51**). The analysis and comparison of the spectral data and optical rotation with those of the bis(methylsulfanyl) derivative led to assigning the same stereostructure for leptosin N_1_ (**52**) [103].

Leptosins M, M_1_, N, and N_1_ (**49**–**52**, Figure 7) were evaluated on the cancer cell growth of the murine P388 lymphocytic leukemia cell line and the HCC disease-oriented panel of 39 human cancer cell lines in the Japanese Foundation for Cancer Research. Significant cytotoxicity against the P388 cell line was found for these natural products, with leptosins N and N_1_ (**51** and **52**, respectively) being almost 10 times more potent. (+)-Leptosin M (**49**) also showed cytotoxic activities against the 39 human cancer cell lines [103]. Moreover, the compound at 10 µg/mL inhibited protein kinases PTK and CaMKIII by 40–70%, as well as topoisomerase II (IC_50_ = 59.1 µM) without affecting topoisomerase I [103]. The pattern of differential toxicity against the cancer cell lines using the COMPARE program suggested an unconventional mode of action for (+)-leptosin M (**49**) [103].

**Figure 8 molecules-27-07585-f008:**
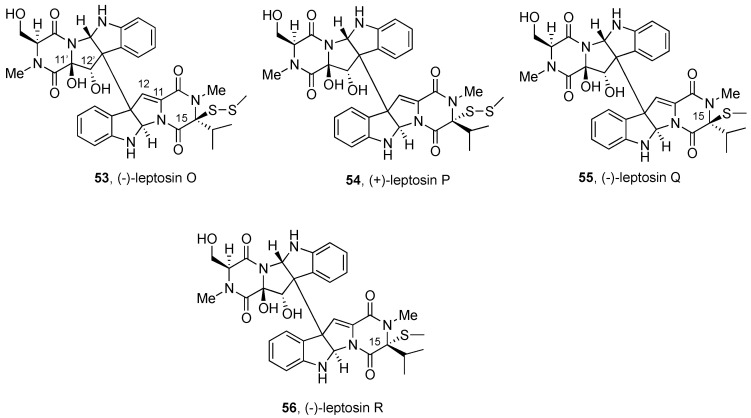
BPI dioxopiperazine alkaloids with sulfur substituents on one of the monomers.

Leptosins O–R (**53**–**56**, respectively, Figure 8), which lack the disulfide bridge of the series, were also isolated from a strain of *Leptosphaeria* sp. OUPS-N80 originally separated from the marine alga *Sargassum tortile* C. Agaroh (Sargassaceae) [104]. An additional analog, termed (−)-leptosin S, is a sulfur-deficient natural product and has not been included.

The structure of (−)-leptosin O (**53**, Figure 8) showed a particular functionality within the series, since it contains a formal acylenamine and a methyldithio substituent. These functionalities were located at the C11=C12 and C15 positions, respectively, on the same unit, as shown after ^1^H-^1^H COSY, HMBC correlations, and DEPT, as well as additional ^1^H-^1^H and ^1^H-^13^C COSY experiments on a triacetate derivative and comparison of the data with those of other family members. The relative configuration was assigned based on NOE correlations, which predicted that the isopropyl-bearing and the acetoxymethyl-dioxopiperazine rings adopt chair and boat conformations, respectively. The same analysis on the series of compounds suggested that (−)-leptosin O (**53**) adopts a B type conformation (Figure 6B) in CDCl_3_. Marfey’s analysis after acid hydrolysis identified L(*S*)-*N*-methylserine as a degradation product, which led to proposing the absolute configuration of (−)-leptosin O (**53**) [104].

(+)-Leptosin P (**54**, Figure 8) generated instead a diacetate derivative under the same conditions, and its NMR spectra were similar to those of the corresponding triacetate of (−)-leptosin O (**53**). NOE correlations of this compound suggested (+)-leptosin P (**54**) to be a stereoisomer of (−)-leptosin O (**53**) at C15, whereas its derivative did not incorporate an acyl group at C12’. Conformational analysis also indicated for (+)-leptosin P (**54**) the B type conformation (Figure 6B) in CDCl_3_ solution. Marfey’s analysis as described above led to proposing its stereostructure with 15*R* configuration [104].

Since the molecular formula for (−)-leptosin Q (**55**, Figure 8) showed one less sulfur atom than that of (−)-leptosin O (**53**), the presence of a methylthio instead of methyldithio substituent was suggested. Accordingly, the triacetate derivative of (−)-leptosin Q (**55**) was identical to that obtained from the reaction of (−)-leptosin O (**53**) with NaBH_4_ and methylation with CH_3_I, followed by Ac_2_O in pyridine, and exists as a B type conformer (Figure 6B) in CDCl_3_ [104].

(−)-Leptosin R (**56**, Figure 8), with the same molecular formula as (−)-leptosin Q (**55**), was proposed to be the stereoisomer at C-15, as confirmed after analysis of the NMR signals of the triacetate derivative, which was also of the B type conformer in CDCl_3_ [104].

(−)-Leptosin O and (+)-leptosin P (**53** and **54**, respectively, Figure 8) exhibited significant cytotoxicity (ED_50_ values of 1.1 and 0.1 µg/mL, respectively, cf. for 5-fluorouracil, ED_50_ = 5.8 × 10^−2^ µg/mL) against cultured murine P388 lymphocytic leukemia cell line [104]. In addition, (−)-leptosin O (**53**) and (−)-leptosin S (**54**) showed moderate cytotoxic activity (mean value of log GI_50_ of −4.01 and −4.01 M, respectively) against the 39 human cancer cell lines (HCC panel) of the Japanese Foundation for Cancer Research [104].

## 3. Biogenesis of BPI-ETPs 

Biogenetic studies of sulfur containing moieties in natural products [105], in particular those from marine organisms [106], as well as comprehensive report on the biosynthesis of pyrrolidinoindoline containing NPs, including BPI-ETPs [107], together with the flavoenzyme-catalyzed formation of disulfide bonds in natural products biosynthesis [108,109], have been previously covered. The biogenetic routes to these natural products are usually found in dedicated biosynthetic gene clusters [31,110,111].

Whereas the incorporation of sulfur to a putative dehydrodioxopiperazine was first proposed to occur immediately following the cyclodipeptide formation, *N*-methylation was proposed to occur after the oxidative cyclization [112]. Methionine, cysteine, and sodium sulfate jointly provide the sulfur connections of thiodioxopiperazines [10]. 

Most of the biosynthetic studies have been carried out on monomeric gliotoxin (**7**, Figure 1B) from *Aspergillus fumigatus*, in which the dioxopiperazine core is assembled by a nonribosomal peptide synthetase [113]. The presence in a fermentation broth of a cyclic dipeptide intermediate bound to glutathione suggested the latter to be the donor of sulfur atoms [114]. Gene knockout experiments revealed that *gli*G was responsible for encoding a glutathione sulfur transferase, namely, GliG, which incorporated the sulfur atom into the DKP framework. Prior to sulfur incorporation, the DKP should undergo an oxidation with the attachment of a hydroxyl group at the Cα-position, for which gliC was identified as the responsible P450 monooxygenase [46,113,114].

The biosynthesis of (−)-gliotoxin (**7**, Figure 2A) in *Aspergillus fumigatus* was elucidated by Hertweck et al., after their discovery of the activation of DKPs by oxygenase GliC and the transfer of glutathione by a dedicated glutathione *S*-transferase, GliC [114,115], which led to the formation of bis(glutathione) adducts **58** from cF_L_-S_L_ (**57**) and lately to the natural product from the dithiol intermediate **59** (Figure 2) [114]. Additional insights were obtained from the large-scale fermentation of an engineered ΔgliI mutant, from which bis(cysteine) conjugates (**60**) were indeed isolated, en route to the dithiol intermediate **61**. The step involving the dual cleavage of the C–S bonds to afford the dithiol prior to disulfide bond formation was also confirmed. The GliI C–S lyase-catalyzed reaction concomitantly forms two thiol groups, in what might reflect the dual action of the GliI homodimer [116].

More recent studies on the biogenesis of pretrichodermamide A (not shown) containing a formal α,β-disulfide (instead of the α,α-disulfide of this series) in *Trichoderma hypoxylon* confirmed that the formation of the disulfide bond to generate the epidithiodioxopiperazine was promoted by an FAD (**63**)-dependent oxidoreductase (Figure 2B). The thiol disulfide oxidases promoted the disulfide bond formation from dithiol **62** to disulfide **64** with substrate and catalytic promiscuities. Not only the disulfide bond of pretrichodermamide A (not shown) (TdaR) but also those of aspirochlorine (not shown) (AclT) and gliotoxin (**7**, Figure 1) (GliT) were generated efficiently (see Figure 2B) [35]. 

The biosynthetic gene cluster (*cha*) of (+)-chaetocin A (**1**, Figure 1) was identified in the producing fungi *Chaetomium virescens* ATCC 26417 through bioinformatic comparison with the BGCs of (−)-gliotoxin (**7**, Figure 1) and sirodesmin (not shown) [117]. Three cytochrome P450 enzymes (ChaB, ChaC, and ChaE) were identified as part of the cha cluster. Although the gene cluster *cha* from *Chaetomium virescens* ATCC 26417 was proposed to generate (+)-chaetocin A (**1**, Figure 3) [117], through radical dimerization, it has not been possible to express ChaE and confirm its activity. Given that detailed information on the tailoring of the glutathione adduct by the γ-glutamylcyclotransferase to generate the DKP disulfide, the timing of sulfuration, the DKP release, and the dimerization, has not yet been clarified, the biogenesis of these natural compounds is still uncertain [117].

The discovery of the gene cluster for the biosynthesis of (+)-verticillin A (**8**, Figure 3) [118] confirmed that *ver*P encoded a NPRS in connection with other genes besides *verA* and *verK*. The disulfide bond formation by oxidation of the dithiols was likely promoted by VerT, similar to GliT, and *verZ* controls the production of the natural product [119]. The biogenesis of (+)-verticillin A (**8**) was abolished by the disruption of *verM* or *verG*, encoding a putative *O*-methyltransferase and a glutathione *S*-methyltransferase, respectively [118,119]. These mutants produced unrelated metabolites termed gliocladiosin A and B (structures not shown). No additional mechanistic details have been reported for the series.

The biosynthetic gene cluster of (+)-verticillin A (**8**) from the *Cordyceps*-colonizing fungus *Clonostachys rogersoniana* (formerly known as *Gliocladium* sp.) isolated from the *Cordycep* fruiting body collected in Tibet Linzhi County (China) has more recently been identified and cloned (Figure 3). Based on the sequence of a nonribosomal peptide synthetase (ChaP), which was predicted to be responsible for (+)-chaetocin A (**1**) biosynthesis in *C. virescens*, the verticillin biosynthetic gene cluster (*ver*) was constructed. Disruption of the nonribosomal peptide synthetase gene *verP* in the *ver* cluster was shown to abolish the production of (+)-verticillin A (**8**) [120]. Disruption of the *O*-methyltransferase gene *verM* on the same species stopped the production of (+)-verticillin A (**8**) and allowed for isolating two cryptic compounds (not shown), which were identified as dipeptides conjugated with macrolides [121].

## 4. Total Synthesis of Tryptophan-Derived BPI-ETP Alkaloids

Prior reviews on the synthesis of sulfur-containing moieties of NPs [105] and on strategies for the total synthesis of bispyrrolidinoindoline dioxopiperazines have been recently published [3,32]. Since, in all reported synthesis, the dimeric structures are constructed starting with a functionalized pyrrolidinoindoline fragment, an introduction to dimer formation follows, preceding the total synthesis of the natural products. 

### 4.1. General Construction of Tryptophan-Derived Bis-Pyrrolidinoindolines 

Movassaghi et al. developed a Co(I)-mediated dimerization of 3a-bromocyclotryptophans **74** or **75**, obtained from **73**. The reaction of DKP **73** with bromine in acetonitrile provided bromopyrrolidinoindolinedioxopiperazine *exo*-**74** and *endo*-**75** in a 44:56 ratio and in a 86% yield. Treatment of bromopyrrolidinoindolinedioxopiperazines with tris(triphenylphosphine)cobalt(I) chloride CoCl(PPh_3_)_3_ (**79**) in acetone was used as a key step for the synthesis of diastereomeric bispyrrolidinoindoline dioxopiperazines (Figure 4B) [122,123]. In all cases, the configuration of the hexahydropyrrolo[2,3-*b*]indole ring fusion was conserved along the dimerization process, and this could be traced back to the configuration of the tryptophan derivative used in the bromocyclization step (Figure 4B).

Our group optimized reaction conditions (one equivalent PPTS, CH_2_Cl_2_, 25 °C) [124] for the diastereoselective bromocyclization reaction of protected D-tryptophan derivative **76** with *N*-bromosuccinimide (NBS) to generate in 93% yield *exo*-**77** and its diastereomer *endo*-**78** in a 94:6 ratio and in 93% yield (Figure 4A), a very high diastereoselectivity justified by DFT-based computations [125].

As indicated above, dimerization of the bromopyrrolidinoindolines **79** upon treatment with CoCl(PPh_3_)_3_ (**79**) generated in 57% yield the bispyrrolidinoindoline **80 [122]**. Moreover, the configuration at C-2/C-2’ in homodimer **80** was known to be efficiently inverted and afforded **81** (93% yield) using lithium hexamethyldisilazide (LiHMDS, four equivalents) at −15 °C in THF and quenching of the corresponding lithium ester enolates with MeOH at −78 °C (Figure 4B) [126].

A bioinspired approach reported by Tadano et al. [127,128] to the series of symmetrical and nonsymmetrical natural products involved the dimerization reaction of tryptophan derivatives in aqueous acidic media. Four dimeric compounds (**83**/**84** and **85**/**86**) with C-3/C-3’ (*exo-* and *endo-*diastereomers) and C-3/C-6’ connections were prepared starting from **82** through the use of either Mn(OAc)_3_ under homogeneous conditions or V_2_O_5_ under heterogeneous conditions [128].

Tu et al. developed a KI-catalyzed dimerization of bromopyrrolidinoindolines **87** with the formation of an intermediate iodine cation, generated by the oxidation of iodide ions (I^−^) with NaBO_3_, to afford the bis-pyrrolidinoinolines **88** and **89**, although in a 1.2:1 diastereomeric ratio (Figure 5B) [129].

The dimerization of protected tryptophan methyl ester (namely, **90**) by a copper-mediated radical reaction (CuCl_2_, DBU, acetonitrile, 0 °C) further optimized the selectivity of the coupling process, since it afforded a mixture of dimeric bis-pyrrolidinoindolines **91/92** in an *exo*/*endo* ratio that varies with the amino acid protecting group [130]. Thus, whereas with *o*-Ns groups the *endo*-**92** was favored (1:3 *exo*/*endo*), the trend was changed to *exo*-**91** (3: 1 *exo*/*endo*) by using a *p*-Ns protecting group in efficient overall yields (based on reacted starting material) of 55% and 63%, respectively (Figure 5C).

Nondimerization approaches involve Movassaghi’s bromocyclization/Friedel-Crafts arylation [131,132] and Reisman’s one-step synthesis of C3-aryl pyrrolidinoindoline dioxopiperazines via Cu-catalyzed arylation of *N*-tosyltryptamines [133,134,135] using bis(mesityl)-α-diamine ligands and diphenyliodonium triflate ((Ph_2_I)OTf). 

### 4.2. Synthesis of (+)-12,12’-Dideoxyverticillin A (***4***)

Movassaghi and coworkers carried out the first total synthesis of (+)-12,12’-dideoxyverticillin A (**4**) through condensation of protected natural amino acids L-Trp and L-Ala prior to the formation of dioxopiperazine **93** (Figure 6) in what was considered as a route that mimics putative biosynthetic steps [136]. Bromocyclization of the cyclic dipeptide **93** to *endo*-**94** and homodimerization promoted by CoCl(PPh_3_)_3_ (**79**) afforded the octacyclic dimer **95** in a moderate yield (46%). In the search for efficient oxidation of the C-Hα position relative to the carbonyl groups of the dioxopiperazines, they faced substrate limitation due to formation of diastereomeric mixtures, partially oxidized, and degradation by-products. Nevertheless, they could carry out their functionalization under mild oxidative conditions since the putative radical enjoys captodative stabilization. Using Py_2_AgMnO_4_ the dimeric derivative **96** (63%) was obtained as the only diastereomer, and its configuration was confirmed through X-ray diffraction analysis. This result was surprising since the four hydroxyl groups are oriented towards the interior of the dimeric structure and, therefore, towards the more sterically hindered regions. A mechanistic proposal was suggested involving the abstraction of the DKP hydrogen by MnO_4_^−^, followed by immediate trapping by the same face of the generated radical with an oxygen atom of the permanganate reagent [136].

Comprehensive experimentation revealed the relevance of the relative and absolute configuration of the DKP rings due to a combination of nonoptimal conformations of the C–H bonds for abstraction and to the sterically disfavored approach of the oxidant from the concave face of the fused system, thus highlighting the importance of the selection of L-amino acids on synthetic planning. The instability of tetrol **96**, under both acidic and basic conditions, led to protecting the hemiaminal oxygen groups as silyl ether derivatives using TBDMSCl in the presence of the nucleophilic chiral catalyst PPY [(*R*)-(+)-4-pyrrolidinopyridinyl(pentamethylcyclopentadienyl)iron] (**97**), which chemoselectively led to diol **98** resulting from the selective protection of the hydroxyl groups at C_15_ and C_15’_. The disulfide bridges were incorporated using potassium trithiocarbonate in place of hydrogen sulfide. Release of the hydroxyl groups in acidic media generated the corresponding acyliminium ions at the C_11_ and C_11’_ positions, and their trapping with K_2_CS_3_ and ensuing intramolecular attack to C_15_ and C_15’_ afforded **99**, with inversion of configuration, in a moderate yield. Lastly, the addition of ethanolamine provided aminetetrathiol **100**, which, under mild oxidative conditions, led to natural product **4** (Figure 5) [136].

### 4.3. Synthesis of (+)-Chaetocin A (***1***) and (+)-Chaetocin C (***3***)

The synthesis of (+)-chaetocin A and (+)-chaetocin C (**1** and **3**, respectively) and analog (+)-12,12´-dideoxychetracin A (**120**, see Scheme 8) was accomplished using a similar strategy (Figure 7) [137]. Thus, dimeric dioxopiperazine **105** was obtained starting from the bromocyclization of precursor **101**, prepared from the methyl esters of *N*-Boc-L-Trp and L-Ser, which selectively afforded the *endo* diastereomer **102**. Kinetic deprotonation and methylation (MeI) at −40 °C led to **103** in 86% yield. Exchange of protecting groups from silyl ether to acetate uneventfully afforded the precursor **104** for the dimerization step. CoCl(PPh_3_)_3_ (**79**)-mediated reductive radical dimerization generated the dimeric dioxopiperazine **105** (49% yield). Exhaustive and selective tetrahydroxylation of **105** with Py_2_AgMnO_4_ in CH_2_Cl_2_ generated **106** in 55% yield. Regio- and stereoselective incorporation of sulfur substituents could be achieved due to the innate reactivity differences between C11 and C15 and their relative environments. Thus, the reaction of tetrol **106** with TFA in H_2_S-saturated nitromethane provided the bisthiohemiaminal with high diastereoselectivity. Addition of isobutyryl chloride afforded the octacyclic dithioisobutyrate **108** in 53% combined yield. Irradiation of a solution of **108** with a black-light phosphor-coated lamp in the presence of 1,4-dimethoxynaphthalene as a photosensitizer and L-ascorbic acid as a reducing agent afforded the desired dithiol **109** (51% yield). The corresponding bis(triphenylmethanedisulfide) **110** was obtained in 90% yield in a single step after the formation of the dithiol. Ionization of the isobutyrates and cyclization with loss of the triphenylmethyl cation efficiently afforded (82% yield) (+)-chaetocin diacetate (**112**). Methanolysis of the acetates upon treatment with Otera’s catalyst (**113**) at 85 °C provided (+)-chaetocin A (**1**) in 92% yield [136].

Using instead chloro(triphenylmethane)disulfane as a sulfenylation agent allowed for generating bis(triphenylmethanetrisulfide) **114** (86% yield) (Figure 8). Although the protocol used for (+)-chaetocin A (**1**) (Figure 7) led to decomposition due to the lower rates of cyclization of larger rings, an alternative procedure involving the protection of the amines **114** as trifluoroacetamides followed by iminium ion formation at C15 allowed the generation of trithiodioxopiperazine **118** in 91% yield (Figure 8). Methanolysis of the acetates and in situ hydrogenolysis afforded (+)-chaetocin C (**3**) in 95% yield [136].

### 4.4. Synthesis of (+)-12,12´-Dideoxychetracin A (***128***)

For the tetrasulfide-containing (+)-12,12´-dideoxychetracin A (**120**), a similar sequence was optimized by, first, the conversion of the dithiol **109** into the bis(triphenylmethanetetrasulfide) **115** using chloro(triphenylmethane)trisulfane (X = SSS) in 80% yield. The best conditions found in this case for the transformation into the natural product involved the conversion of diamine **115** into bisformamide **117** and rapid removal of the acetyl and formyl groups of **119** by acid-catalyzed methanolysis, which generated (+)-12,12´-dideoxychetracin A (**120**) in 52% yield (Figure 8) [136].

Interestingly, whereas the epidisulfides and epitrisulfides proved to be single conformers in solution on the normal NMR time scale at room temperature, (+)-chaetocin C (**3**) and its diacetate (**121**) were found to exist in solution as a mixture of conformers (for example, in a 1:1.5:2.4 ratio at 90 °C in toluene-*d_6_*, with an estimated activation energy for interconversion of ca. 23 kcal/mol in solutions of 1,3,5-trimethylbenzene-*d_12_*) [136].

**Scheme 8 molecules-27-07585-sch008:**
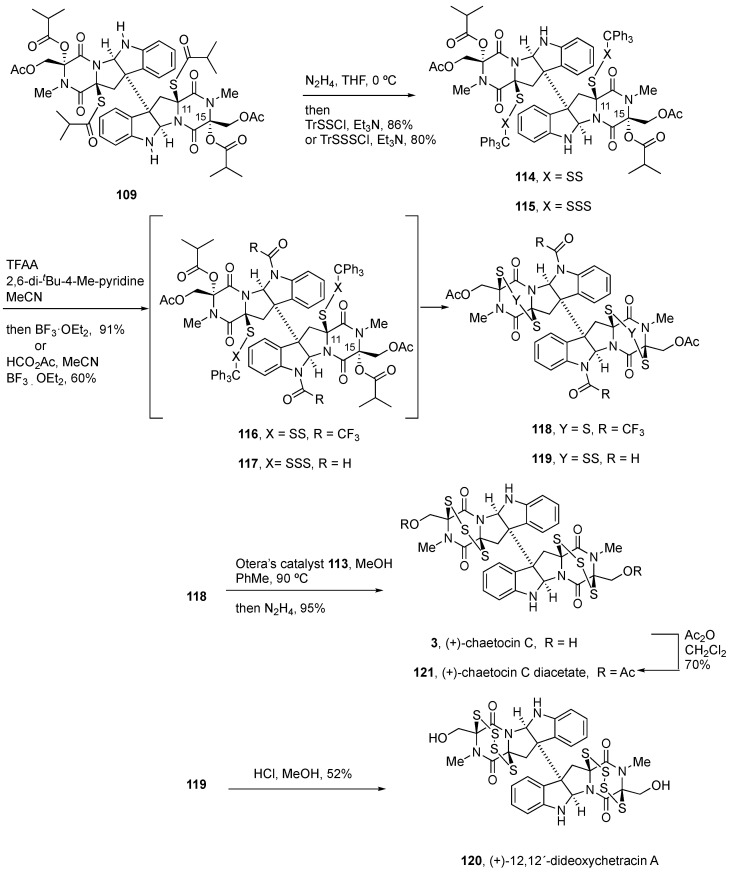
Synthesis of (+)-chaetocin C and (+)-chaetocin C diacetate (**3** and **121**, respectively) and (+)-12,12’-dideoxychetracin A (**120**) [137].

### 4.5. Synthesis of (+)-Chaetocin A (***1***)

Sodeoka and coworkers [45,138,139], using a related strategy, completed the first total synthesis of (+)-chaetocin A (**1**) (Figure 9). Starting from the condensation of un-natural amino acids D-Trp and D-Ser, the resulting DKP **122** was subjected to bromocyclization using NBS, affording with high diastereoselectivity the *exo* adduct **123**, likely due to the presence of the *N*-methyl substituent on the DKP. Radical generation from tetracyclic monomer **123** with AIBN led to producing degradation. However, using instead V-70 (**124**) as a radical initiator (Figure 9), piperazine diol **126** was obtained in an almost quantitative yield from a putative tribromide intermediate **125**. Thus, the treatment of **123** with a phosphate-buffered solution led to the corresponding diol **126** with the suggested putative configuration. Despite the instability of the diol monomer, the radical dimerization afforded the corresponding dimeric tetrol **127** as a single isomer. Lastly, the treatment of **127** with H_2_S in the presence of BF_3_·OEt_2_ as Lewis acid gave rise to the addition of the thiols, by the mediation of the presumably acyliminium ion intermediates, through the external face of the dimeric structure and afforded (+)-chaetocin A (**1**). Remarkably, four substitution reactions (OH to SH), the deprotection of four protecting groups sensitive to acidic media, and the formation of two S–S bonds were achieved in the same step [45,138,139].

## 5. Biological Activities of Natural Product-Inspired (Dithio)dioxopiperazines

Early studies on structurally related (+)-chetomin (**128**, Figure 9), which is produced by certain strains of filamentous fungi, such as *Chaetomium cocliodes* and *Chaetomium seminudum* [140], indicated that the natural product was able to disrupt hypoxia-inducible factor 1 (HIF1) activity through direct targeting of the interactions between the α subunit and p300 coactivator or its orthologue, the CREB-binding protein (CBP). As a result, the compound blocked the transactivation of the hypoxia-inducible gene (HIF1α) expression [44,141]. Moreover, (+)-chetomin (**128**) was also reported to induce coagulative necrosis, anemia, and leucocytosis in animal experiments [141].

To shed some light on the activities of (+)-chetomin (**128**, Figure 9) and related natural products, a series of simple DKPs-containing disulfides (**129**) and the corresponding reduced analogs with thiol functionalities were prepared (Figure 9). The first group of synthetic DKPs with disulfide bridges was shown to also disrupt C-TAD and CH1 binding, which suggested that a simple core structure of the dithio-DKP was sufficient to elicit this activity. Dithiol **130** proved also to be active, but methylsulfide analogue (**132**) was inactive, and therefore, the presence of a thiol or a disulfide functionality was concluded to be required for activity [44].

In a cancer cell-based model, the addition of zinc eliminated the antiproliferative effects of both natural and synthetic ETPs (**128–132**) and drastically increased the percentage of viable cells. Addition of zinc restored the cell’s production of VEGF in hypoxia. It was concluded that the disruption of zinc binding may be one of the potential generic mechanisms of epidithiodioxopiperazine (ETP) action [44].

In addition, a transcriptional antagonist with a dimeric epidithiodioxopiperazine skeleton has been designed to selectively disrupt the interaction of HIF1α with p300/CBP coactivators and downregulate the expression of hypoxia-inducible genes [142,143]. Interestingly, the simple ETP derivative **134** (PS-ETP-1) (Figure 10) obtained from the condensation of Pro and Ser, followed by oxidation to the diol and reaction with H_2_S in BF_3_**^.^**OEt_2_ to **133**, followed by oxidation with I_2_, as well as the trisulfide **135**, showed comparable G9a inhibitory activities (IC_50_ around 5 µM) and reduced cytotoxicity [66,144].

(+)-Chaetocin A (**1**) was first isolated from *Chaetomium* species fungi and reported as an antitumor and antifungal natural product [19]. Additional inhibitory activities against the epigenetic enzyme lysine-specific histone methyltransferase (HMTs) were reported [63,69].

The enantiomer (−)-chaetocin (*ent*-**1**, see Figure 10) was also synthesized following the same sequence described in Figure 9 [45,139], and found to exhibit more potent apoptosis-inducing activity through caspase-8/caspase-3 activation than natural (+)-chaetocin (**1)** [145]. In addition to the natural product and its enantiomer (−)-chaetocin (*ent*-**1**), the *S*-deficient analogs (not shown), protected derivatives, tetrathiol **137**, and monomeric disulfide **136** were also synthesized [144]. The simplified monomer **136** exhibited slightly stronger G9a-inhibitory activity than parent (−)-chaetocin (*ent*-**1**), which suggested that the dimeric structure was not required. However, its enantiomer, namely, *ent*-**136**, showed weaker activity [144]. Inhibitory activities against protein lysine methyl transferase H3K9 HMT G9a [66] of sulfur-deficient chaetocin and of the enantiomers of these compounds (not shown) revealed the importance of the disulfide bond for HMT inhibition and the lack of enantiomeric discrimination [144]. Moreover, these compounds exhibited lower cytotoxicity than monomer **136** and no inhibition of TrxR, in contrast with (+)-chaetocin (**1**), which suggested that the dimeric indole structure and the disulfide bond of the natural product are concomitantly required in order to densely occupy the binding site of TrxR [144].

Although (+)-chaetocin (**1**) was reported [63] to be a specific inhibitor of the SU(VAR)3–9 class of histone lysine methyl transferases (HKMTs) [66], more recent studies demonstrated that it is a nonspecific inhibitor of the epigenetic enzyme [70,71,146]. Desulfuration experiments of (+)-chaetocin (**1**), produced by *Chaetomium virescens* var. *thielavioideum* cultured in solid complete media, using triphenylphosphine in dichloromethane at ambient temperature allowed for generating the ring-contracted analogue **139** in 93% yield [56,70,71]. Computations at the ωB97XD/6-311++G(d,p) level using Gaussian 09 [102], estimation of the optical rotation compared with those of the diastereomers, and simulated ECD curves suggested that the reaction took place with retention of configuration. Therefore, careful structural analysis of these samples and mechanistic studies [147] further confirmed that only the structurally unique ETP core was required for inhibition, which, moreover, is time dependent and irreversible (in the absence of DTT), resulting, at least in part, in protein denaturation. Similar considerations should be applied to the other alkaloids of the series that were reported as G9A inhibitors.

Additional activities reported for the series involve the induction of oxidative stress and cytotoxicity to various cell types [10,148,149,150,151]. Upon reduction of the disulfide bridge to the corresponding dithiol by cellular reductants, the redox cycling activities between both species were shown to generate reactive oxygen species, which induces cell death. Additional mechanisms involving ROS-mediated processes [152] and ETP-binding protein-mediated mechanisms [153] have also been proposed.

In contrast to (+)-gliocladicillin C (C77) (**16**, Figure 2), synthetic disulfide-cleavage product termed C77-5 (**17**, Figure 2) did not induce caspase-dependent apoptosis in human tumor cells (HeLa, T24, A549, HCT116, MCF7, and SW480), reactive oxygen species production, and PARP cleavage [90], but further enhanced the autophagic flux compared with the natural product, thus confirming that the disulfide functionality in (+)-gliocladicillin C (C77) (**16**) was required for the induction of caspase-dependent apoptosis but not for the activation of autophagy.

Verticillin derivatives **140** and **141** (Figure 11) were generated by semisynthesis from the natural products and their activities screened against cancer cell lines [154]. The core structure of the natural products (+)-verticillin A (**8**, Figure 11) and (+)-verticillin H (**23**, Figure 11), isolated from cultures of *Clonostachys rogersoniana* (fungal strain MSX595553), was exploited with the aim of synthesizing derivatives of the latter through selective formation in one of the monomeric units of both straight-chain and more sterically encumbered branched-chain esters, as well as carbonate, carbamate, and sulfonamide derivatives (**141a–i**). For (+)-verticillin A (**8**), only the succinate derivative (**140**) was prepared (Figure 11) [154].

The antiproliferative effects on human melanoma cancer cell (MDA-MB-435), human breast cancer cells (MDA-MB-231), and human ovarian cancer cells (OVCAR3) were determined for these analogs. Acylated verticillin H derivatives (**141a–d**) retained or slightly improved the activity of the parent compound in both cellular and targeted assays, with IC_50_ values from 7 to 229 nM (*cf*. for verticillin H (**23**), IC_50_ values of 31, 229, and 44 nM for MDA-MB-231, OCVAR3, and MDA-MB-435, respectively). Therefore, the substitution of the core structure appears to enhance the physicochemical properties, including cell permeability, of the parent natural compounds [154].

Additional studies on ETPs addressed the design of simple analogs that could maintain or improve the desired activities. It was already known that ETPs accumulate in cells in a glutathione-dependent manner even under well-oxygenated conditions and exist in the cytoplasm almost exclusively in its reduced form [153]. Based on the conformationally averaged distance (ca. 10 Å) estimated in (+)-chetomin (**128**, Figure 10), dimeric structure **142** was designed, with the aim of probing the role of the disulfide bridge on each monomer [142]. SPR experiments in the presence of DTT revealed that compound **142** (Figure 12), similar to (+)-chetomin (**128**), bound directly to the GST-tagged CH1 domain of human p300. Both enantiomers and the *meso* form were found to interact with similar binding affinities. The K_D_ values were dependent on the presence of DTT, with a 36-fold decrease of the oxidized form towards the p300 CH1 domain, which indicated that a reduction step was required in order to obtain high-binding affinity [142]. Analog **142** showed antitumor efficacy in a breast carcinoma model with rapid regression of tumor growth that lasted for up to 14 days, thus suggesting the potential of these compounds to overcome hypoxia-induced tumor growth and resistance [143].

These transcriptional antagonists with a dimeric epidithiodioxopiperazine skeleton have been shown to selectively disrupt the interaction of HIF1α with p300/CBP coactivators and downregulate the expression of hypoxia-inducible genes [142,143]. Both (+)-chetomin (**128**, Figure 9) and analog **142** (Figure 12) were able to alter the global fold of the p300 CH1 domain, disrupting the HIF-1α-p300 CH1 complex and inhibiting HIF-1-induced transcription. The significant reduction of promoter activity and efficient reduction of the expression of HIF-1α inducible genes responsible for promoting angiogenesis and metastasis were considered of therapeutic interest. Moreover, synthetic analog **143** (Figure 12) showed lower toxicity than (+)-chetomin (**128**), whereas monomer **144** exhibited lower values (31% inhibition), which led to the conclusion that the dimeric structure was also important for binding the receptor and promoting the biological activity [142].

Inspired in the structure of natural ETPs, a novel skeleton was designed that maintains only three of the rings present in natural products (+)-chetomin (**128**, Figure 9) and (+)-chaetocin A (**1**, Figure 1). With the general structure of 1,4-dioxohexahydro-6*H*-3,8a-epidithiopyrrolo[1,2*a*]alkylpyrazines, 19 molecules were synthesized and tested for biological activity on human prostate cancer (DU145) and melanoma (A2508) cell lines. The most potent of the series, namely, **145** (Figure 12A), showed IC_50_ values of 0.1 and 0.06 µM, respectively, in these cell lines (cf., (+)-chaetocin A **1**, 0.073 and 0.061 µM in these cell types) and additionally exhibited nanomolar activity against both solid (Huh-7 liver) and blood (MV4-11 AML) cancer cell lines (IC_50_ values of 1.8 and 3.3 nM, respectively). The synthetic compound **145** significantly suppressed tumor growth in xenograft tumor models of A2058 human melanoma and A549 lung cancer by IP injection or oral administration, with no signs of toxicity [155].

Comprehensive studies by Movassaghi and coworkers on the synthesis and biological evaluation of a collection (more than 60 derivatives) of ETP-containing compounds allowed the generation of structure–activity relationship profiles regarding the structural modifications on N1, C3, C17, and C11/C15 positions of the general skeleton (Figure 12). Evaluation of the antitumor activity was carried out in five cancer cell lines, U-937 (histiocytic lymphoma), HeLa (cervical carcinoma), H460 (lung carcinoma), 786-O (renal carcinoma), and MCF-7 (breast carcinoma), after a 72 h exposure [131]. Potency was associated with the presence of the disulfide bond and correlated with the steric environment at C3, the dimeric ETP alkaloids being the most potent with (sub)nanomolar IC_50_ values against the cancer cell lines, in particular, compound **146**, a formal bis-phenylsulfonyl derivative of (+)-12,12’-dideoxyverticillin A (**4**, Figure 1) (with IC_50_ values of 0.18 ± 0.06 nM by MTS for U-937 cells and 0.09 ± 0.06 nM, 1.53 ± 0.85 nM, 1.55 ± 0.77 nM, and 1.65 ± 0.51 nM, respectively, for HeLa, H460, 786-O, and MCF-7 cells) by SRB (sulforhodamine B). Although substitutions of the dimers at N1 and C17 could be used for compound optimization, the monomers were also shown to have more optimal pharmacokinetic properties [131].

Further studies were directed towards the design and chemical synthesis of analogs with functional linkers. The most promising of the series were those with an alkyl azide group at C3, N1, and N14 positions (**147**–**150**, Figure 12). Sarcosine-derived analogues **147**, **149**, and **150** were found to maintain the described cytotoxicity of parent **146**, whereas **148** showed slightly reduced activities against HeLa, A549 (alveolar adenocarcinoma), and DU 145 (prostate carcinoma) cell lines [156].

## 6. Summary and Conclusions

The structural diversity and variety of homo- and heterodimeric polycyclic tryptophan-derived alkaloids is another evidence of the unique ability of fungi to biosynthesize complex skeletons even with dissimilar dimeric structures that become challenging targets for synthesis. As such, only homo- and heterodimeric BPI-ETPs with the same configuration of the units have been synthesized using the Co(I)-promoted heterodimerization of bromopyrrolidinoindolines. However, the tactic was found to be of little use (16% of the desired heterodimer, together with the homodimer in the same statistical amounts, and mixtures of disproportionation and side by-products were obtained) for the synthesis of model nonsymmetrical BPI dioxopiperazines with different configurations at the C3-C3’ connection [123]. Luckily, heterodimeric and nonsymmetrical BPI dioxopiperazines have been synthesized [123], presumably through radical coupling processes, using the photoexpulsion of dinitrogen from mixed diazenes in a solvent cage, although no total synthesis of these alkaloids has yet been reported. 

The most relevant bioactivities of the series are dependent on the presence of di(poly)sulfide units, since the dimeric structures are not required in order to elicit these actions. Thus, monomers have been designed and synthesized, and they displayed optimal pharmacokinetic properties. 

Once again, inspiration from nature has provided key skeletons as starting points to design simple analogs that could maintain or improve the desired activities and, therefore, deserve further optimization for their development as small-molecule drugs.

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
