# Peer review of "Bispyrrolidinoindoline Epi(poly)thiodioxopiperazines (BPI-ETPs) and Simplified Mimetics: Structural Characterization, Bioactivities, and Total Synthesis†"

_molecules, 2022, doi:10.3390/molecules27217585_

Round 1
Reviewer 1 Report
The article titled "Bispyrrolidinoindoline Epi(poly)thiodioxopiperazines 2 (BPI-ETPs) and Simplified Mimetics: Structural Characterization-3, Bioactivities and Total Synthesis" covers aspects of biological activity, synthesis and biosynthesis of a certain class of compounds, namely pyrrolidinoindolinedioxopiperazines containing bridged sulfur bonds. The review is consistent, first covering various classes of representatives of such compounds, and then the narrative proceeds directly to the synthesis, which, due to the complexity of the structures, occupies a fairly small volume of text. The English language is understandable and does not require editing.
There are non-critical repetitions in the text, which I would not delete, since they are necessary for the development of thought. I'll give just a couple of examples:
sentence
"(+)-chaetocin A (1) and some functionally-related analogues revealed that only the epidithiodioxopiperazine core was required for the reported inhibition of the SU(VAR)3-9 class of HKMTs.[63]"
and
"Although (+)-chaetocin (1) was reported[63] to be a specific inhibitor of SU(VAR)3-9 class of histone lysine methyl transferases (HKMTs),[66] more recent studies 1246 demonstrated that it is a non-specific inhibitor of the epigenetic enzyme.[70, 71, 146]"
sentence
"Melinacidin IV (also known as (+)-12,12’-dihydroxychaetocin 13; first named 361 (+)-11,11’-dihydroxychaetocin)[19] was also isolated from Westerdykella reniformis sp. nov, together with (+)-chetracin B or HDN-1 (28, Figure 4).[85]"
and
"(+)-Chetracin B or HDN-1 (28)[68])[95] was also isolated, together with (+)-12,12’-dihydroxychaetocin or melinacidin IV (13, Figure 2),[73] from the algicolous fungus Westerdykella reniformis obtained from Prince Edward Island, Canada,[85]"
Again, I do not consider it necessary to exclude these passages from the text.
In general, I want to congratulate the authors on a good job and can recommend the article for publication.
I just have a couple of notes:
1. Scheme 1 is too far from the discussion in the text.
2. 11a,11a’-dihydroxychaetocin) The opening parenthesis is missing.
3. In sentence: “Movassaghi et al. developed a Co(I)-mediated dimerization of 3a-bromocyclotryptophans 74 and 75” In Scheme 4, there is no compound 75, only endo and exo 74. Consider editing or re-numbering.
4. In sentence: “The reaction of bromopyrrolidinoindolinedioxopiperazine 73 with CoCl(PPh3)2 in acetone was used it as key step for the synthesis of diastereomeric bispyrrolidinoindoline dioxopiperazines.” Obviously, bromopyrrolidinoindolinedioxopiperazine should be 74, moreover, this reaction is not shown in Scheme 4, which is not clear.
5. In sentence: “Moreover, the configuration at C-2/C-2’ in homodimer 80 was known to be efficiently inverted and afforded 79” compound 79 should be 81.
6. In Scheme 6, it makes no sense to introduce R2 for 96 since there is no substituent variability there.
Author Response
I just have a couple of notes:
- Scheme 1 is too far from the discussion in the text.
It has been moved to page 2
- 11a,11a’-dihydroxychaetocin) The opening parenthesis is missing.
It was included
Melinacidin IV (also known as (+)-12a,12a’-dihydroxychaetocin 13; first named (+)-11a,11a’-dihydroxychaetocin)
- In sentence: “Movassaghi et al. developed a Co(I)-mediated dimerization of 3a-bromocyclotryptophans 74 and 75” In Scheme 4, there is no compound 75, only endo and exo 74. Consider editing or re-numbering.
It has been corrected and the numbers assigned accordingly.
Movassaghi et al. developed a Co(I)-mediated dimerization of 3a-bromocyclotryptophans 74 or 75, themselves obtained from 73. The reaction of DKP 73 with bromine in acetonitrile provided bromopyrrolidinoindolinedioxopiperazine exo-74 and endo-75 in a 44:56 ratio and in a 86% yield. Treatment of bromopyrrolidinoindolinedioxopiperazines with CoCl(PPh3)2 (79) in acetone was used it as key step for the synthesis of diastereomeric bispyrrolidinoindoline dioxopiperazines (Scheme 4B).[122, 123] In all cases, the configuration of the hexahydropyrrolo[2,3-b]indole ring fusion was conserved along the dimerization process and this could be traced back to the configuration of the tryptophan derivative used in the bromocyclization step (Scheme 4B).
- In sentence: “The reaction of bromopyrrolidinoindolinedioxopiperazine 73 with CoCl(PPh3)2 in acetone was used it as key step for the synthesis of diastereomeric bispyrrolidinoindoline dioxopiperazines.” Obviously, bromopyrrolidinoindolinedioxopiperazine should be 74, moreover, this reaction is not shown in Scheme 4, which is not clear.
Corrected to:
As indicated above, dimerization of the bromopyrrolidinoindolines 79 upon treatment with CoCl(PPh3)2 (79) generated in 57% yield the bispyrrolidinoindoline 80.[122] Moreover, the configuration at C-2/C-2’ in homodimer 80 was known to be efficiently inverted and afforded 81 (93% yield) using lithium hexamethyldisilazide (LiHMDS, 4 equivalents) at -15 °C in THF and quenching of the corresponding lithium ester enolates with MeOH at -78 °C (Scheme 4B).[126]
- In sentence: “Moreover, the configuration at C-2/C-2’ in homodimer 80 was known to be efficiently inverted and afforded 79” compound 79 should be 81.
Corrected as indicated above
- In Scheme 6, it makes no sense to introduce R2 for 96 since there is no substituent variability there.
Corrected
Reviewer 2 Report
Martínez et al. submitted title, "Bispyrrolidinoindoline Epi(poly)thiodioxopiperazines (BPI-ETPs) and Simplified Mimetics: Structural Characterization, Bioactivities and Total Synthesis" is a well-written review.
The main strength of the paper is in the structural illustration. Authors covered Tryptophan-Derived 99 BPI-Etps Alkaloids (BPI-ETPs Alkaloids, Chaetocin and Analogues, Gliocladines and Chetracins, Leptosins and Preussiadins)
The authors put focused on chemical characterization with two-dimensional NMR spectroscopy and X-ray crystallography for structural elucidation of these natural compounds.
In my opinion, this article is a perfect fit for the issue, "In Honor of Prof. Dr. Fernanda Borges’s Contribution"
However, there are some subjective issues regarding the drawing of some of the chemical structures where I could see the overlapping of heteroatom, but because of the complexity of these structures, these issues are expected.
Secondly, the authors could make a table enclosing the alkaloids and related properties.
I recommend this paper for this issue.
Author Response
Reviewer 2
However, there are some subjective issues regarding the drawing of some of the chemical structures where I could see the overlapping of heteroatom, but because of the complexity of these structures, these issues are expected.
Drawings have been revised
Secondly, the authors could make a table enclosing the alkaloids and related properties.
We considered first to present the family of alkaloids with a Table, but for the discussion of the structure-activity relationships we finally decided to discuss each family in global analysis.
I recommend this paper for this issue.